# Near Neutral Selectionist Theories (NNST) for SARS-CoV-2 suggested by the substitution-mutation ratio (*c/μ*) analysis

Chun Wu[1,2]*, Nicholas J. Paradis[1]

**1** Department of Chemistry and Biochemistry, Rowan University, Glassboro, New Jersey, United States of America, **2** Department of Biological & Biomedical Sciences, Rowan University, Glassboro, New Jersey, United States of America

☉ These authors contributed equally to this work.
* wuc@rowan.edu

## Abstract

A definitive test to measure genome-wide fitness effects of any nucleotide mutation, including translated regions (*TRs*) and untranslated regions (*UTRs*), is essential to help resolve the decades-long neutralist–selectionist debate regarding mutation-mediated species evolution. The precise boundary, composition, and abundance of nearly neutral mutations remain disputed, highlighting the need for a rigorous framework supported by empirical sequence data. Our substitution–mutation rate ratio test (*c/μ*) might provide such a framework. *c/μ* compares the ratio of how often mutations fix into the population (substitution rate, *c*) with their expected arrival (mutation rate, *μ*), which classifies each mutation type (*c/μ > 1*: adaptive; *c/μ = 1*: neutral; *c/μ < 1*: deleterious). We previously showed that SARS-CoV-2 exhibits L-shaped distributions of fitness effects (*DFEs*) and a strict molecular clock, and mutation type proportions consistent only with the Near-Neutral Balanced Selectionist Theory (*NNBST*) and not with conventional molecular evolution theories. However, a theoretical explanation for incidences of non-strict clock behavior in several SARS-CoV-2 segments are not formalized. Here, we extended *c/μ* analysis to 49 segments of SARS-CoV-2 (26 TRs, 12 UTRs, and 10 transcriptional regulatory sequences (*TRSs*)) and provide formal, mathematical frameworks for our *NNBST* and Near-Neutral Unbalanced Selectionist Theory (*NNUST*) to explain non-strict clock behavior. All 49 segments displayed L-shaped *DFEs*: 24 segments (mostly *TRs*) supported molecular clocks and balanced effects of near neutral mutations, consistent with *NNBST*; meanwhile, 25 segments (mostly *UTRs/TRSs*) did not support molecular clocks or balancing of near neutral mutations, consistent with *NNUST*. Numerous violations of Selectionist Theory (*ST*), Kimura's Neutral Theory (*KNT*), and Ohta's Nearly Neutral Theory (*ONNT*) were observed, but none for *NNBST* or *NNUST*. Together, these results support a unified Near-Neutral Selectionist Theory (*NNST*), combining neutral and selectionist perspectives to better explain the molecular evolution of SARS-CoV-2.

**Data availability statement:** All relevant data are within the paper and its Supporting information files.

**Funding:** This work was supported by the National Science Foundation (NSF ACI-1429467, NSF RUI-1904797, and ACCESS/BIO230145) and the New Jersey Health Foundation (PC 76-24). Computational resources were provided by the Pittsburgh Supercomputing Center through allocation PSCA170090P. The funders had no role in study design, data collection and analysis, decision to publish, or preparation of the manuscript.

**Competing interests:** The authors have declared that no competing interests exist.

## Introduction

All theories of molecular evolution agree that most new mutations in a genome are strongly deleterious ($f^- \gg f_0 \; or \; f_0' \; or \; f^+$: the fraction of strongly deleterious ($f^-$), strictly neutral ($f_0$), nearly neutral ($f_0' = f_0^- + f_0 + f_0^+$) including weakly deleterious ($f_0^-$) and weakly beneficial ($f_0^+$), and strongly beneficial ($f^+$) mutations in the genome, respectively) and thus will be purified by negative selection, but these theories differ in their hypotheses on the proportion of the strictly neutral, nearly neutral and strongly beneficial mutations that have fixed in a population (i.e., the proportion of the different types of substitutions) [1].

The Selectionist (Neo-Darwinian) Theory/*ST* [2,3] states that the fraction of overall substitutions are strongly beneficial (i.e., $f^+ \gg f_0 \; or \; f_0'$), leading to a time-dependent substitution rate due to positive selection (**Eq. 1a** $c = c^+ f^+$; $c^+$ : the time-dependent substitution rate of strongly beneficial mutations). In contrast, Kimura's Neutral Theory (*KNT*) [4–7] argues that most substitutions are strictly neutral mutations (i.e., $f_0 \gg f_0' \; or \; f^+$), leading to a time-independent substitution rate (A.K.A. strict molecular clock) that's due to strictly neutral selection acting on it (**Eq. 2a** $c = \mu f_0$, $c$, the substitution rate; $\mu$: the time-independent mutation rate [7]). *KNT* suggests the "junk" part of the genome, including Un-Translated region (*UTR*) such as introns, pseudogenes, and all low-complexity genomic regions and synonymous sites in a translated region (*TR*), are likely under strictly neutral evolution with an evolution rate approaching the mutation rate [7]. Ohta's Nearly Neutral Theory/*ONNT* [8,9] extends *KNT*'s perspective by acknowledging the importance of nearly neutral mutations that mostly have weakly deleterious effects ($f_0^+ \approx 0, f_0^- + f_0 \gg f^+$) besides the strictly neutral mutations [1]. *ONNT* predicts a negative correlation between the substitution rate and the effective population size. Consequently, the change of the effective population size changes over time leads to a time-dependent substitution rate (**Eq. 3** $c = \mu (f_0 + q (1 - f_0))$; $q$ : time-dependent proportion of deleterious mutations that reach fixation) [1].

Despite extensive study, establishing clear boundaries between neutral and non-neutral mutations remains a nontrivial challenge with both semantic and technical implications, as such classifications are critical for empirically evaluating evolutionary theories [8,10,11]. Traditional definitions of near-neutrality, such as those by Ohta (see Table 1), are explicitly population-size–dependent (i.e. $N_e$ -dependent) and classify mutations based on the condition ($|N_e s| \leq 2$ or $|N_e \sigma| \leq 1$, s: selection coefficient; σ: standard deviation of the selection coefficient), where the neutral status of a mutation relative is compared to the assumed effective population size. Furthermore, the composition and relative abundance of nearly neutral mutations remain contentious. Whereas Ohta emphasized the role of slightly deleterious substitutions [8,9], Gillespie modeled symmetric weak effects [12] and Sella and Hirsh demonstrated detailed balance [13] (see Table 2). Ohta originally proposed that most mutations are weakly deleterious ($d = 2\mu t(f_o + q(1 - f_o))$) [8,9]. In contrast, Gillespie proposed that the fractions of weakly deleterious and weakly beneficial mutations, under non-equilibrium and under equilibrium conditions, are symmetrical to each other (i.e., 50/50; ($\pi_o^- = \pi_o^+$ and $f_o^- = f_o^+$) [12]. Ohta later revised her proposal to include both

**Table 1. Methods for defining nearly neutral mutation boundaries within a _c/μ_ Distribution of Fitness Effects.** The nearly neutral mutation boundaries encompass the weak negative boundary (_WNB_) and weak positive boundary (_WPB_) _c/μ_ values of nearly neutral mutations for a genomic segment, and the _WNB_ and _WPB_ _c/μ_ values determined from our SARS-CoV-2 empirical sequence data.

| Method | Method description | WNB and WPB _c/μ_ values for SARS-CoV-2 |
|---|---|---|
| Ohta 2002 (ref 8) | *$\|N_e s\| < z$, z between 0 and 2 Arbitrarily, Population-Size-dependent, define WPB $c/\mu \approx 2*$ and WNB $\approx 0.5$ (assuming symmetry) | WNB: 0.52 WPB: 2.02 |
| Ohta, 1972 (ref 10) Ohta & Tachida 1990 (ref 11) | *$\|N_e \sigma\| < 1$, $\sigma$ denotes the standard deviation of the selection coefficient. Population-Size-dependent, Theoretical | N/A |
| This study | Uses fraction of nearly neutral sites defined by segment-based _c/μ_ to define WNB; WPB _c/μ_ is defined from WNB assuming symmetry, random generic drift is confirmed by constant substitution rate. Empirical, outcome-based, | WNB: 0.05 WPB: 19.01 |

\*$c/\mu = N_e * P_{fix}$, where $P_{fix} \approx 2s$. If: $\frac{c}{\mu} = N_e * 2s$, _then_ $s = \frac{\frac{c}{u}}{2N_e} \leq \frac{1}{N_e}$. Furthermore, $\frac{c}{u} \leq \frac{2N_e}{N_e}$, therefore $\frac{c}{u} \leq 2$ (_WPB_) and $\frac{c}{u} \geq 0.5$ (_WNB_) assuming symmetry.

**Table 2. Theory summary concerning the fixation of weakly neutral mutations.** Equation description, presence of molecular clock, detailed balance condition, evidence to support theory, as described prior (Ohta, Sella, Gillespie) in comparison to the fixation of nearly neutral mutations as described in the Near-Neutral Selectionist Theory proposed in this study.

| Theory | Ohta | Sella | Gillespie | This study |
|---|---|---|---|---|
| Key argument concerning fixation of weakly neutral mutations | Most are weakly deleterious | Detailed balance condition for the fixation and proportions of weakly deleterious and weakly beneficial mutations | Equal proportions of weakly deleterious and weakly beneficial mutations | Asymmetric proportions of weakly deleterious and weakly beneficial mutations, with different balancing of their substitution rates and fractions |
| Equation | $d = 2\mu t(f_o + q(1-f_o))$ | $\pi_o^- c_o^- = \pi_o^+ c_o^+$ | $\pi_o^- = \pi_o^+$ $f_o^- = f_o^+$ $\frac{1}{\sqrt{2\pi\sigma^2}} \int_{s\sigma-1/2N}^{s\sigma} e^{-y^2/2\sigma^2} dy$ $= \frac{1}{\sqrt{2\pi}} \int_{s-1/\alpha}^{s} e^{-z^2/2} dz$ | $(\mu - c_0^-)f_0^- = (c_0^+ - \mu)f_0^+$ |
| Molecular clock | No | Yes | No | Yes (if balanced) No (if not balanced) |
| Detailed balance condition | No | Yes | No | Yes |
| Evidence to support theory | Mathematical framework, simulation data | Mathematical framework, simulation data | Mathematical framework, simulation data (House-of-cards) | Mathematical framework, empirical sequence data |
| Reference | 14 | 13 | 12 | |

*Gillespie: s: Mean selection coefficient; π: equilibrium fraction of weakly deleterious mutation; N: effective population size; y: absolute fitness of genotype y; z: absolute fitness of genotype z; α: strength of selection.

weakly deleterious and a smaller, asymmetric fraction of weakly beneficial substitutions (i.e., $f_o^- \gg f_o^+$) [14]. Alternatively, Sella and Hirsh proposed that the observed fractions of weakly deleterious and weakly beneficial mutations are assumed to be at their equilibrium distributions, following the detailed balanced condition [13]; in other words, the product of the fraction of weakly deleterious mutations with their substitution rate under weakly deleterious selection is equivalent to the product of the fraction of weakly beneficial mutations with their substitution rate under weakly beneficial selection, under the stationary state ($\pi_o^- c_o^- = \pi_o^+ c_o^+$, $\pi_o^-$ and $\pi_o^+$: equilibrium fractions of weakly deleterious and weakly beneficial mutations; $c_o^-$ and $c_o^+$: substitution rates of weakly deleterious and weakly beneficial mutations) [13]. In Ohta, Sella and Hirsh and Gillespie's cases, their proposals are based on constructing a mathematical framework that is dependent on theoretical simulation data (i.e., Gillespie and house-of-cards model) probably due to the lack of empirical sequence data available at

their initial proposals. However, a mathematical framework for testing the fractions of different mutation types must also be supported by substitution mutation data from empirical sequence analysis as opposed to just simulation data, when available. These theories, including *ST*, *KNT* and *ONNT*, could not be evaluated on the basis of dated, empirical sequence data, unless all mutations in *TR* and *UTR* could be accurately classified by their fitness effects into distinct types and their proportions could be accurately quantified by a rigorous method with sufficient empirical genomic sequence data, thus leading to a continuing 50-year "neutralist-selectionist" debate [1,15–17].

Genomic sequence evolution, shaped by mutation, genetic drift, and natural selection, can be modeled as a continuous-time Markov process in which nucleotide or codon changes occur over time [18]. Two major frameworks are commonly used to study the mode and strength of selection in protein-coding sequences [19,20]. The *Ka/Ks* framework compares non-synonymous substitution rates (*Ka*) to synonymous substitution rates (*Ks*) within *TR*, interpreting *Ka/Ks* > 1 as positive selection, *Ka/Ks* < 1 as purifying selection, and *Ka/Ks* ≈ 1 as neutral selection. While effective for studying selection pressures acting in *TR* [21,22], this method assumes synonymous mutations are effectively neutral ($Ks = \mu$ = constant) [23] and does not apply to *UTRs*, despite non-natural selection reportedly acting on synonymous sites in *TRs* and *UTRs* [2,24]. In contrast, the mutation-selection (MutSel) framework estimates the effective population-scaled selection coefficient ($S = 2N_e s$) [25] for each nucleotide site, offering insight into the inferred fitness effects of mutations [26]. However, the MutSel framework assumes a static fitness landscape and is computationally intensive to solve.

To overcome these limitations, we developed the unified $c/\mu$ framework, which uses the ratio of the substitution rate (*c*) to the approximated mutation rate ($\mu$) to estimate the time- and site-specific selection pressures from empirical genomic sequence data (see Figure in S1 Fig) [20]. The $c/\mu$ ratio compares how often mutations arise in each genomic segment ($\mu$) to how often they are fixed in the population (*c*) over evolution time. Each genomic site bearing a $c/\mu$ value can be broadly classified into three different mutation classes ($c/\mu$ > 1 as positive selection due to a higher fixation rate, $c/\mu$ < 1 as purifying selection due to a lower fixation rate and $c/\mu$ = 1 as neutral selection due to *c* having the same fixation rate as its expectation rate from $\mu$). The frequency and probability of each mutation class can provide the combined forces of deleterious, nearly neutral and beneficial selection acting within each genomic segment. The $c/\mu$ framework, which is grounded firmly in population genetics theory under Wright, Fisher and Haldane, bridges the mechanistic detail of the scaled selection fitness change due to mutation from the MutSel framework with the population-level outcomes of mutation frequency from the *Ka/Ks* framework (see Figure B in S1 Fig). Unlike *Ka/Ks* methods, the $c/\mu$ method is not required to assume mutation models, enabling unbiased analysis of mutation effects in nucleic acid evolution [20] and is applicable to both nucleotide sites in *TRs* and *UTRs* to obtain their complete fitness change in addition to protein fitness change (see Figure B (right) in S1 Fig). Moreover, $c/\mu$ supports high-resolution assessments of substitution timelines, Site Substitution-Mutation Rate Ratio Spectra (*SSMRRS*) and distributions of fitness effects (*DFEs*), which are key to understanding how mutation, genetic drift and natural selection combined shape genome evolution [17]. In addition, the $c/\mu$ *DFE* utilizes a novel approach based on empirical sequence data to more rigorously define the nearly neutral selection boundaries and quantify the nearly neutral mutations in genomic segments, instead of wholly relying on arbitrary boundary cutoffs.

Unlike Ohta, Sella and Hirsh and Gillespie's prediction-based approaches (see Table 2) which pre-define the *DFE* boundaries to quantify the precise fractions of non-neutral, neutral and nearly neutral mutations in a genome, our $c/\mu$ framework operates as an outcome-based framework that defines the *DFE* boundaries and estimate mutations under near-neutrality using observed $c/\mu$ values of each genomic segment, bypassing the need to specify $N_e$ and instead estimating selection strength directly from evolutionary outcomes. This approach yields an $N_e$–independent and reproducible criterion for classifying neutral, nearly neutral, and non-neutral selected mutations. Unlike Ohta's boundaries, which are fixed without reference to the observed distribution of $c/\mu$ within the dataset, our method defines the weak negative boundary (*WNB*) and weak positive boundary (*WPB*) based on genomic segment-specific $c/\mu$ values, thereby capturing the full range of nearly neutral mutations that shape evolutionary dynamics. Moreover, the formalized $c/\mu$ framework demonstrates that an asymmetric proportion of weakly deleterious mutations (likely in higher abundance) and weakly

beneficial mutations (likely in lower abundance) exists within a genome and are either in balance with one another $(\mu - \bar{c_0})f_0^- = (c_0^+ - \mu)f_0^+$ or not $(\mu - \bar{c_0})f_0^- \neq (c_0^+ - \mu)f_0^+$ (see Table 2).

To demonstrate the advantages of our $c/\mu$ framework, we used it to analyze SARS-CoV-2's empirical genomic sequence data compiled by NextStrain, comprising three independent datasets totaling 11,198 genome sequences collected over the first 19-months of the COVID-19 pandemic [27]. Two notable features are inconsistent with those predicted by the conventional molecular evolution theories (*KNT*, *ONNT* and *ST*) in our first study [17]. First, the genomic substitution rate (*GSR*) of SARS-CoV-2, derived from its substitution timeline, is a time-independent constant and a strict molecular clock, despite fluctuations in weekly infection cases and increasing vaccination rates [17]. This temporal substitution behavior is inconsistent with the time-dependent substitution expectation by *ONNT* and *ST*, but is consistent with the time-independent substitution expectation of *KNT*. But this consistency with *KNT* ends there: More strikingly, the $c/\mu$ *SSM-RRS* for the SARS-CoV-2 genome across 29,903 nucleotides, to quantify site specific mutation fitness effects, revealed an L-shaped $c/\mu$ *DFE*, which is inconsistent with *KNT*, *ONNT* and *ST* because of the unexpected fractions of different mutation types (86% strongly deleterious, 7% weakly deleterious, 0.5% strictly neutral, 3% weakly beneficial, 4% strongly beneficial) obtained from the L-shaped $c/\mu$ *DFE*. We thus proposed the Near-Neutral Balanced Selectionist Theory (*NNBST*), which acknowledges the non-negligible contribution of nearly neutral mutations and strongly beneficial mutations, while simultaneously challenging the core tenet of *KNT* and *ONNT* by suggesting that strictly neutral mutations are rare (0.5%) and do not contribute significantly to generating the strict molecular clock, and that weakly beneficial mutations are negligible (3%). Instead, the strict molecular clock is due to balancing between weakly beneficial and weakly deleterious mutations for the SARS-CoV-2's genome and its composing genes [7] ($c = f_o'\mu$, see methods section) as opposed to the genetic drifting mechanism proposed by *KNT* ($c = f_o\mu$ and $f_o \leq 1$).

Second, while we reported the fractions of mutation types for the SARS-CoV-2 genome, those for its comprising *TR*s, *UTR*s and Transcription Regulation Sequences (*TRS*s) have not been quantified. Empirical sequence data, a prerequisite for computing the fractions of mutation types, was unavailable at the initial proposals of *ST*, *KNT* and *ONNT*, thus their evaluation for an entire genome and their *TRs*/*UTRs*/*TRSs* have not been performed for any species. Some signatures of their fractions at the segment level were gleaned from our prior experiments. We previously inferred the overall selection fitness change inclusive of the fitness change due to nucleic acid and protein sequence change in the SARS-CoV-2 *TR*s and *UTR*s, indicating they are at least under weakly to strongly deleterious selection (i.e., $c/\mu$, $Ka/Ks$ and $Ks/\mu < 1$), which contrasts with the neutrality assumptions of synonymous substitution rates in *TR* and *UTR*s by $Ka/Ks$ tests under Nei-Gojobori (NG), Li-Wu-Luo (LWL), Pamilo-Bianchi-Li (PBL) and Yang-Goldman Maximum Likelihood (ML) methods [20]. This strongly suggests that the use of $Ka/Ks$ is necessary, but not sufficient to obtain the overall fitness effects from protein fitness change alone because of the observed non-neutral synonymous substitution rates due to nucleic acid fitness change, and thereby, $c/\mu$ offers the generalized approach to obtaining the overall fitness change of genomic segments, which already incorporates the $Ka/Ks$ metric. Moreover, some codon and nucleotide sites in SARS-CoV-2 *TR*s and *UTR*s were revealed to be under strong beneficial selection (>69 nonsynonymous mutation sites), supported by their predicted $c/\mu$ values ($c/\mu > 3$) being consistent with the experimental characterization of increased infectivity, immune escape and drug resistance, to name a few [28], indicating strongly beneficial mutations are rare but not negligible. Although *NNBST* can adequately describe the time-independent *GSR* and some gene segments of SARS-CoV-2 instead of *KNT*, several gene substitution rates did not exhibit a strict molecular clock for this virus. This indicates the strict molecular clock is not as ubiquitous as is assumed and that *NNBST* is not followed for all genomic segments [17]. At that time, a formal mathematical framework for explaining the absence of strict molecular clocks in genomic segments was not proposed. Thereby, if the fractions of mutation types for genomic segments without strict molecular clocks cannot be explained by *ST* or *ONNT*, they may likely be described by a novel unbalanced selection mechanism, Nearly-Neutral Unbalanced Selectionist Theory (*NNUST*).

In this study, we extend our *c/μ* analysis to the 26 *TRs*, 12 *UTRs* and 10 *TRSs* of the SARS-CoV-2 genome, against empirical sequence data, to determine their strict molecular clock feature (or absence of), quantify the proportions of different mutation types and evaluate them against the five theories of molecular evolution. We also rigorously construct the mathematical framework for *NNBST* and *NNUST* to provide a novel interpretation of the molecular clock with an L-shaped *c/μ DFE* for a genomic segment. Our analysis suggests each genomic segment most likely follows *NNBST* and *NNUST* instead of the three conventional theories of molecular evolution. For the ease of the reader, see Table 3 for definitions of major symbols and terms mentioned throughout this paper.

## Results

### Relatively uniform mutation profile in the SARS-CoV-2 genome justifies the constant mutation rate assumption as a first-order approximation

To verify the mutation rate (*μ*) is a time-independent constant, we summarized the mutation rate per site per year (*S/N/Y*) and mutation rate per site per replication cycle (*S/N/R*) values from reported cell-based and cell-free experiments in our previous study [20]. The viral polymerase assays report an elongation rate of ~50–100 nucleotides/second [29],

**Table 3. Definitions of major symbols.**

| $n_c/Nc$ | Census population of a virus |
|---|---|
| *Ne* | Replicated population of a virus before the selection step |
| *G* | Number of genomes |
| *N* | Number of nucleotide sites in the sequence |
| *t* | Collection date of a genome |
| *Δt* | $Δt = t_{max} - t_{min}$ |
| $m_{ij}$ | Substitution count matrix (Time, Position) |
| $P_{mut}$ | Mutation probability |
| $P_{fix}$ | Fixation/Substitution* probability |
| *μ* | Number of mutations per nucleotide site per unit time/Mutation rate |
| *c* | Number of substitutions per nucleotide site per unit time/Substitution rate |
| *c/μ* | Substitution/mutation rate ratio/Relative substitution rate |
| *SSMRRS* | Site Substitution-Mutation Rate Ratio Spectrum |
| *DFE* | Distribution of Fitness Effects/Distribution of Relative Substitution Rate |
| *WNB* | Weak negative boundary for DFE |
| *WPB* | Weak positive boundary for DFE |
| $c^-, c_0^-, \mu, c_0^+, and\ c^+$ | Mean substitution rate of strongly deleterious, weakly deleterious, strictly neutral, weakly beneficial and strongly beneficial sites |
| $f^-,\ f_0^-, f_0, f_0^+\ and\ f^+$ | Fraction of strongly deleterious, weakly deleterious, strictly neutral, weakly beneficial and strongly beneficial sites |
| *TR* | Translated Region |
| *UTR* | UnTranslated Region |
| *GSRM* | Genomic Substitution Rate Model |
| *SSRM* | Segment Substitution Rate Model |
| *ST* | Selectionist (Neo-Darwinian) Theory |
| *KNT* | Kimura's Neutral Theory |
| *ONNT* | Ohta's Nearly Neutral Theory |
| *NNST* | Near-Neutral Selectionist Theory |
| *NNBST* | Near-Neutral Balanced Selectionist Theory |
| *NNUST* | Near-Neutral Unbalanced Selectionist Theory |

leading to about 5–10 minutes for replicating one SARS-CoV-2 genome under cell-free conditions. In cell culture (human WD-PNECs, Caco2, Calu3, A549-ACE2, hNECs and monkey Vero E6 cells), the complete SARS-CoV-2 replication cycle takes on the order of 2–12 hours. Each of the mutation studies reported mutations after the final cell passaging event (see Table in S1 Table). Polymerase misincorporation assays give ~$10^{-3}$ errors per site per reaction, which is 3–4 orders of magnitude higher than typical cell-based mutation rates, because there is no proofreading or repair mechanism present. In vitro cell-based mutation rates span an $S/N/R$ of ~$0.5–15 \times 10^{-6}$, varying by cell type, MOI, passage length, and viral genotype (proofreading intact vs. ExoN1). Our in vivo mutation rate ($\mu$), obtained from the effective neutral substitution rate method [20], reports an $S/N/Y$ value of $4.30 \times 10^{-3}$, which is very consistent with the in vitro "year" $S/N/Y$ extrapolations of most studies (~$10^{-3}$), since most of these studies were conducted on the order of days. This provides strong evidence to support our assumption that the mutation rate is indeed time-independent for SARS-CoV-2. To further validate our site-independent mutation rate assumption for a genome, we plotted 1) the reported cell-based nucleotide mutations and 2) the percent nucleotide substitution rates from our genomic sequence datasets in the SARS-CoV-2 genome (see Fig 1 and Table in S1 Table). First, the cell-based nucleotide mutations (black lines) are distributed across most genes but exhibit reduced density in NSP3 (3CL-like protease), NSP11 (RdRp), NSP12 (helicase), NSP13 (exoribonuclease), NSP14 (endoribonuclease) and NSP15 (methyltransferase). This clearly shows a lack of experimental data spanning sufficiently long timeframes to show a more uniform mutation distribution, but also might suggest the strong deleterious selection acting on more critical genes to conserve their functionality even at the cellular level, while other genes such as the spike glycoprotein show increased mutation density, potentially from their less deleterious fitness impact. Second, our phylogenetic data of mutations fixed in the human population after selection (orange lines) show a more uniform mutation distribution across the genome except in NSP13, NSP14 and NSP15; most nucleotide sites exhibit a relatively low substitution rate (<10%), but a few sites exhibit relatively higher substitution rates in NSP3, NSP11 and N (>25%), indicating varying substitution rates at different nucleotide sites. The difference in the mutation distribution is likely from the phylogenetic sequence data being much denser and spanning a much longer time scale. The lack of mutations in NSP13, NSP14 and NSP15, across the cell-based mutations and our phylogenetic data, seemingly agree that these genes are under quite strong deleterious selection, since they are involved in RNA proofreading, host mRNA degradation and RNA

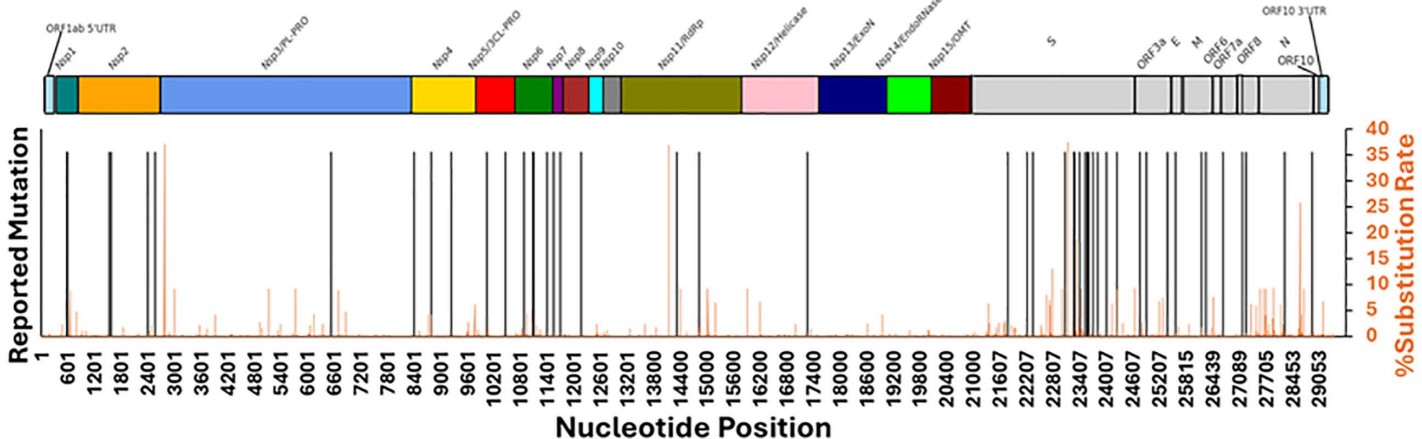

**Fig 1. Investigation of the site-independent mutation rate across each nucleotide site in the SARS-CoV-2 genome.** The mutation landscape of the SARS-CoV-2 genome in the short time frame (black lines, mutation occurrence following cell passaging) and longer time frame after selection (orange lines, position-based percent substitution rate of each nucleotide site from SARS-CoV-2 genomic sequence data analyzed in this study). See Table of S1 Table for the nucleotide mutations compiled from our previous study.

methylation to evade the host immune response, critical for maintaining the virus's presence in the host cell. From these data, our constant mutation rate assumption as a first-order approximation is likely satisfied for SARS-CoV-2.

### A Strict Molecular clock was observed for 24 out of 49 genomic segments within *UTR* and *TR* of SARS-CoV-2

The time-based total substitution rate (*c*) and coefficient of determination ($R^2$) values were tabulated for each segment (see Table of S2 Table for *UTR/TRS* and Table of S3 Table for *TR*). $R^2$ is a statistical measure for indicating the molecular clock feature ($R^2 \geq 0.6000$) or not ($R^2 < 0.6000$) for a segment (see Methods section for details) [20]. The substitution time-lines of Orf1ab 5'UTR, genome, All-TR, All-UTR, All-TRS, most genes in *TR* and nonstructural proteins (*NSPs*) (see Table of S3 Table for the 24 segments) demonstrate a strict molecular clock, whereas most *UTRs/TRSs*, some genes in *TR* and *NSPs* (see Table of S2 Table for the 25 segments) do not demonstrate a strict molecular clock (see Fig 2 for all segments combined; see Figures in S6–S9 Figs for the individual substitution timeline diagrams). Only 3 out of 22 *UTRs/TRSs* exhibited strict molecular clocks, specifically Orf1ab 5'UTR ($R^2 = 0.9392$), All-UTR ($R^2 = 0.7167$) and All-TRS ($R^2 = 0.6299$), whereas the remaining *UTRs* and *TRSs* did not exhibit strict molecular clocks ($R^2 = -0.2110$ to $0.5577$), with E TRS-B and M TRS-B being entirely conserved ($R^2$ undefined) (see Table in S2 Table). In contrast, 20 out of 25 *TRs* exhibited strict molecular clocks. Besides the genome ($R^2 = 0.9957$) and All-TR ($R^2 = 0.9805$), Orf1ab ($R^2 = 0.9854$), N ($R^2 = 0.9497$), S ($R^2 = 0.8711$), M ($R^2 = 0.7902$), E ($R^2 = 0.7201$), Orf8 ($R^2 = 0.6135$), Orf3a ($R^2 = 0.6116$), NSP1–4 ($R^2 = 0.6555$ to $0.9801$), NSP6–13 ($R^2 = 0.6755$ to $0.9507$) and NSP15 ($R^2 = 0.6361$) exhibited strict molecular clocks, whereas NSP5 ($R^2 = 0.5965$), NSP14 ($R^2 = 0.4841$), Orf7a ($R^2 = 0.4031$), Orf10 ($R^2 = 0.3557$), NSP7 ($R^2 = 0.3018$) and Orf6 ($R^2 = 0.0000$) do not exhibit strict molecular clocks (see Table in S3 Table). The 24 genomic segments exhibiting strict molecular clocks will be

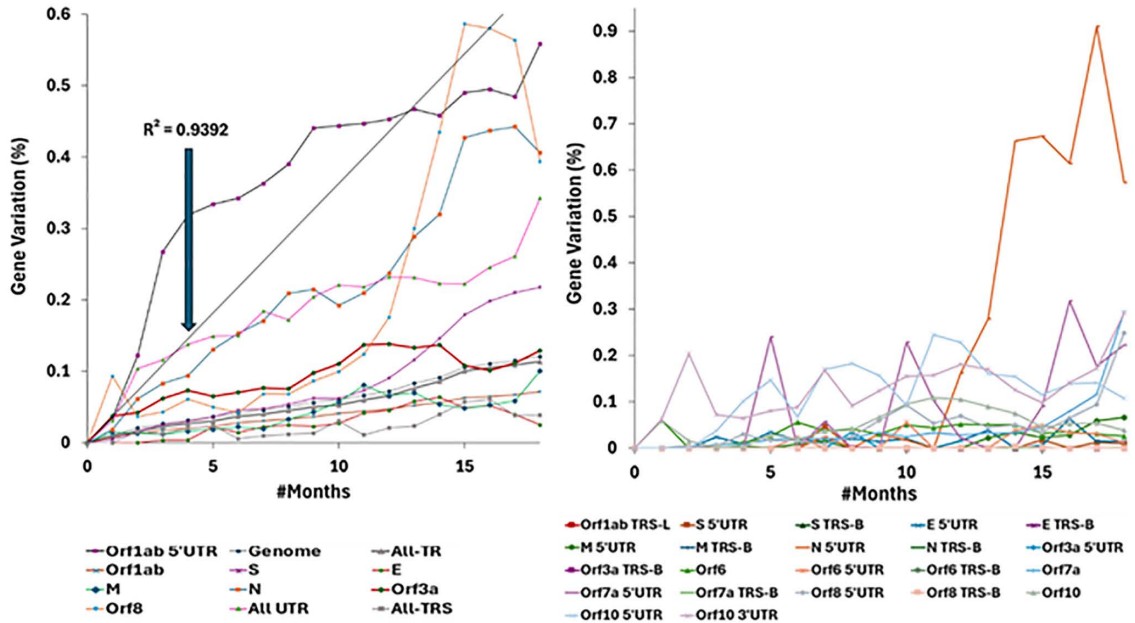

**Fig 2. The varying presence of the molecular clock feature in SARS-CoV-2 genomic segments.** The total percent nucleotide substitution variation (*c*) for segments exhibiting strict molecular clock (A) or no strict molecular clock (B) over evolution time, averaged over the three SARS-CoV-2 datasets. The time-based substitution rate for Orf1ab 5'UTR obtained from its timeline and coefficient of determination ($R^2 = 0.9392$) was defined as the fundamental genomic mutation rate ($\mu = c$ of Orf1ab 5'UTR). See Figures of S6 and S8 Figs for the individual timelines of segments not exhibiting strict molecular clock and Figures of S7 and S9 Figs for the individual timelines of segments exhibiting strict molecular clock.

evaluated against *KNT* and *NNBST*, whereas the 25 genomic segments not exhibiting strict molecular clocks will be evaluated against *ST*, *ONNT* and *NNUST*.

## High consistency between time-based and position-based *c/μ* supports the validation of the strict molecular clock

The consistency between time-based and position-based *c/μ* values (absolute, percent differences) were assessed for the *UTRs*/*TRSs* (see Table of S4 Table) and genes and *NSPs* in *TR* (see Table of S5 Table) to verify the accuracy of the strict molecular clock assumption in the time-based approach. Please see the methods section for the detailed calculations of the time-based and position-based substitution rates. The genomic segments exhibiting strict molecular clocks should exhibit good consistency in their time-based and position-based *c/μ* values; otherwise, lack of strict molecular clocks should produce more deviated time-based and position-based *c/μ* values. The *UTRs*/*TRSs* exhibited larger *c/μ* differences where they exhibited strict molecular clocks (see Table of S4 Table, 0.03 to 0.18, or 1.74% to 8.11%) and those not exhibiting strict molecular clocks (see Table of S4 Table, 0.00 to 0.71, or 0.00% to 55.56%). In contrast, the genes and *NSPs* in *TR* displayed smaller *c/μ* differences for those exhibiting strict molecular clocks (see Table of S5 Table, 0.00 to 0.07, or 0.00% to 12.50%) and larger *c/μ* differences for those not exhibiting strict molecular clocks (see Table of S4 Table, 0.01 to 0.54, or 0.53% to 38.57%). Besides some large percentage differences being observed (e.g., Orf6, Orf7a, Orf10, Orf1ab TRS-L, Orf3a TRS-B, etc.), 21 out of 25 *TRs*, All-UTR, All-TRS and Orf1ab 5'UTR exhibited strict molecular clocks, indicating their selection type assignment by time-based *c/μ* is consistent with the position-based *c/μ* values. The remaining four genes in *TR* (Orf7a, Orf10, NSP5 and NSP7) and remaining *UTRs*/*TRSs* should use the position-based *c/μ* for their accurate selection type assignment.

## Sub-genetic balancing condition between *NSP* substitution rates explains strict molecular clock feature of SARS-CoV-2 Orf1ab

The balanced condition under *NNBST* provides a reasonable explanation for the strict molecular clock observed for SARS-CoV-2 *GSR* [17] and can be extended to explain the strict molecular clock for the Orf1ab substitution rate comprising NSP1–15. To demonstrate this, the timelines for NSP1–15 and Orf1ab were plotted together to show which *NSP* is under weaker or stronger deleterious selection, using the timeline of Orf1ab as a reference point ($c/μ = 0.11$ and $R^2 = 0.9854$) (see Figure of S10 Fig). With respect to the *GSR*, NSP2–3, NSP6, NSP9 and NSP11 likely undergo weaker purifying selection ($c/μ > 0.11$), whereas NSP1, NSP4–5, NSP7–8, NSP10, NSP12–15 likely undergo stronger purifying selection ($c/μ < 0.11$). However, relative to Orf1ab, NSP2–3, NSP6, NSP9 and NSP11 likely undergo beneficial selection and NSP1, NSP4–5, NSP7–8, NSP10, NSP12–15 likely undergo purifying selection. The faster and slower substitution rates of NSP1–15 due to weak beneficial and weak negative selection can balance out and produce the Orf1ab substitution rate, similarly as the individual segment substitution rates of *TR* and *UTR* produce the *GSR*. Hence, *NNBST* can also explain the seemingly time-independent total nucleotide substitution rate of Orf1ab.

## L-shaped discrete *c/μ* DFE suggests *NNST* explains the evolution of SARS-CoV-2

The discrete *c/μ* *DFE*s were L-shaped for the SARS-CoV-2 genome and each *TR*, *UTR* and *TRS* (see Figure of S11–S14 Figs; see Methods section for how the *c/μ* DFE was constructed). The most probable *c/μ* value ($c/μ = 0$) indicating conserved nucleotide sites occur for each segment; as *c/μ* increases, nucleotide sites under weakly neutral selection (weakly deleterious and weakly beneficial selection) and strongly beneficial selection become increasingly rare in a monotonic pattern. This L-shape *DFE* indicates most nucleotide sites within the genome and each segment are not under strict neutral selection. We thus computed the fraction of nucleotide sites under strict neutral selection ($f_0$) for each segment using the discrete *c/μ* DFE. The results are summarized in Tables 4 and 5. The 24 genomic segments exhibiting strict molecular clock all exhibited a small $f_0$ (see Table 4, genome: 0.03% to All-TRS: 1.64%), whereas the 25 genomic segments without strict molecular clock exhibited a wider range for $f_0$ (see Table 5,

**Table 4. Relative rate and mutation fraction values for molecular clock SARS-CoV-2 genomic segments.** Time-based $c/\mu$, $R2$ and percent abundance at three selection type boundaries, the relative percent abundance of sites under five different selection types, proportion ratios and evaluation of *KNT*, *ONNT* and *ST* for the genome and each segment exhibiting strict molecular clock features of SARS-CoV-2 over 19 months using the approximated lower and upper boundaries of $\mu$.

| Seg (NT length) | $f_o = c/\mu$ | c $R^2$ | WNB (c/μ) | NB (c/μ) | WPB (c/μ) | f (SN) | $f_o$ (WN) | $f_o$ (N) | $f_o^+$ (WP) | $f^+$ (SP) | $f_o':f_o$ (NN:N) | $f_o:f_o^+$ (WN:WP) | $f_o':f_o^+ + f^+$ (WN:P) | KNT | ONNT | ST |
|---|---|---|---|---|---|---|---|---|---|---|---|---|---|---|---|---|
| Genome (29,903) | 0.07-0.18 | 0.9957 | 0.05-0.07 | 1.01-1.02 | 14.04-19.34 | 84.21-92.12 | 7.19-14.28 | 0-0.03 | 0.56-1.33 | 0.1-0.19 | 576.7-1858.4 | 5.24-12.66 | 9.43-10.71 | 3,4,5 | NA | NA |
| Orf1ab (21,291) | 0.04-0.11 | 0.9805 | 0.05-0.09 | 1.01-1.01 | 10.8-19.34 | 95.4-98.29 | 0.72-4.27 | 0-0.02 | 0.32-0.93 | 0-0.06 | 468.4-3916 | 9.29-13.18 | 0.73-13.18 | 3,4,5 | NA | NA |
| All-TR (29,133) | 0.06-0.17 | 0.9805 | 0.05-0.07 | 1.01-1.01 | 14.04-19.34 | 85.11-93.2 | 6.14-13.44 | 0-0.03 | 0.54-1.25 | 0.1-0.2 | 586.56-1600 | 5.54-11.26 | 9.31-9.42 | 3,4,5 | NA | NA |
| Nsp3 (5,388) | 0.05-0.14 | 0.8517 | 0.07-0.07 | 1-1.01 | 12.76-14.51 | 85.9-94 | 5.63-13.18 | 0-0.02 | 0.35-0.93 | 0-0 | 668.37-1750 | 9.05-15.66 | 14.25-15.66 | 3,4,5 | NA | NA |
| Nsp12 (1,801) | 0.03-0.08 | 0.9169 | 0.12-0.14 | 0.99-1.01 | 7.02-8.29 | 91.68-96.61 | 3.21-7.1 | 0-0.02 | 0.16-1.17 | 0-0.05 | 366-384.64 | 10.59-19.33 | 5.82-19.33 | 3,4,5 | NA | NA |
| **Orf1ab 5'UTR(265)** | **0.41-1** | **0.9392** | **0.01-0.02** | **1.01-1.01** | **58.14-140.43** | **34.72-63.77** | **35.09-63.4** | **0-0.13** | **1.13-1.89** | **0-0** | **9408-9408** | **31-31** | **31-31** | 3,4,5 | NA | NA |
| N (1,260) | 0.27-0.66 | 0.7902 | 0.02-0.02 | 1.01-1.02 | 46.81-58.14 | 45.64-70.47 | 28.33-51.11 | 0.01-0.08 | 1.19-3.25 | 0-0 | 831.6-2232 | 1.24-23.8 | 15.71-23.8 | 3,4,5 | NA | NA |
| Nsp11 (2,794) | 0.07-0.18 | 0.7922 | 0.03-0.05 | 1.01-1.02 | 19.34-28.08 | 80.79-91.98 | 7.79-18.71 | 0-0.01 | 0.21-0.5 | 0-0 | 3355.2-13216 | 5.99-36.33 | 36.33-37.34 | 3,4,5 | NA | NA |
| S (3,822) | 0.11-0.28 | 0.9854 | 0.03-0.03 | 1.01-1.01 | 28.08-29.02 | 74.05-88.72 | 10.43-24.05 | 0.01-0.03 | 0.83-1.91 | 0-0 | 802.62-861.99 | 2.63-12.46 | 12.46-12.59 | 3,4,5 | NA | NA |
| Nsp2 (1,912) | 0.04-0.12 | 0.6735 | 0.11-0.12 | 1.01-1.01 | 8.29-8.77 | 88.56-94.77 | 4.91-10.14 | 0-0.02 | 0.31-1.31 | 0-0 | 524.98-2750 | 9.52-15.66 | 7.76-15.66 | 3,5 | NA | NA |
| Nsp8 (592) | 0.01-0.04 | 0.6016 | 0.15-0.19 | 1.01-1.08 | 5.28-6.38 | 95.46-97.97 | 2.02-3.87 | 0-0.04 | 0-0.67 | 0-0 | NA | NA | NA | 3,5 | NA | NA |
| Nsp9 (337) | 0.04-0.12 | 0.8214 | 0.1-0.12 | 1.01-1.02 | 8.26-9.67 | 88.5-94.69 | 4.12-8.85 | 0-0.07 | 1.17-2.66 | 0-0 | 172.89-963 | 3.5-31.02 | 3.33-3.5 | 3,4,5 | NA | NA |
| Nsp10 (415) | 0.01-0.03 | 0.7971 | 0.18-0.28 | 1.01-1.17 | 3.63-5.4 | 96.64-98.56 | 1.43-3.12 | 0-0.01 | 0-0.24 | 0-0 | NA | NA | NA | 3,5 | NA | NA |
| M (669) | 0.05-0.12 | 0.7201 | 0.09-0.1 | 1.01-1.04 | 9.36-11.6 | 87.44-94.76 | 4.78-11.51 | 0-0.02 | 0.44-1.05 | 0-0 | 642.24-1732.5 | 9.84-10.66 | 10.66-11 | 3,4,5 | NA | NA |
| E (228) | 0.03-0.08 | 0.8711 | 0.07-0.12 | 1.01-1.02 | 7.8-14.51 | 91.67-96.49 | 3.07-7.46 | 0-0.02 | 0.43-0.44 | 0-0.44 | 501.6-1696 | 7-9.5 | 7-8.5 | 3,5 | NA | NA |
| All-UTR (771) | 0.21-0.52 | 0.6361 | 0.02-0.06 | 1-1.01 | 2.34-58.04 | 33.98-76.52 | 22.56-63.17 | 0-0.07 | 0.64-2.85 | 0-0.25 | 715.1-3848.5 | 2.15-34.8 | 22.14-24.85 | 3,4,5 | NA | NA |
| Nsp13 (1,579) | 0.02-0.07 | 0.9507 | 0.14-0.14 | 1.01-1.06 | 6.68-7.25 | 92.73-96.83 | 2.9-6.39 | 0-0.03 | 0.25-0.82 | 0-0.06 | 265.61-1150 | 8.1-11.5 | 7.21-11.5 | 3,5 | NA | NA |
| Nsp1 (538) | 0.03-0.1 | 0.6135 | 0.12-0.12 | 0.99-1.01 | 7.8-8.29 | 90.19-95.92 | 3.33-8.7 | 0-0.03 | 0.74-0.93 | 0-0.18 | 378-957 | 4.5-10.6 | 4.5-7.83 | 3, 5 | NA | NA |
| Nsp4 (1,498) | 0.03-0.09 | 0.9801 | 0.12-0.12 | 1.01-1.01 | 8.26-8.29 | 91.67-96.26 | 3.46-7.33 | 0-0.02 | 0.26-0.93 | 0-0.07 | 540-1512 | 10.93-13 | 7.33-13 | 3,4,5 | NA | NA |
| Nsp15 (892) | 0.02-0.06 | 0.6755 | 0.14-0.18 | 1.01-1.02 | 5.4-7.25 | 93.88-97.12 | 2.51-4.92 | 0-0.05 | 0.35-1.2 | 0-0 | 112.59-708 | 7-14.06 | 4.1-7 | 3,5 | NA | NA |
| All-TRS (61) | 0.02-0.07 | 0.7167 | 0.33-0.42 | 0.82-1.01 | 3.05-70.21 | 93.22-96.72 | 3.27-3.5 | 0-0.13 | 0-3.28 | 0-0 | NA | NA | NA | 3,5 | NA | NA |
| Orf8 (366) | 0.28-0.68 | 0.6116 | 0.02-0.02 | 1.01-1.1 | 46.81-58.04 | 36.34-66.12 | 31.96-60.38 | 0.01-0.03 | 1.91-3.28 | 0-0 | 1694.66-2613.24 | 8.99-16.71 | 16.71-18.41 | 3,4,5 | NA | NA |
| Orf3a (828) | 0.1-0.24 | 0.9497 | 0.1-0.12 | 1.01-1.03 | 8.26-9.67 | 76.81-90.21 | 9.17-21.01 | 0.03-0.04 | 0.6-2.17 | 0-0 | 324-596.16 | 5.62-15.2 | 9.67-15.2 | 3,4,5 | NA | NA |
| Nsp6 (868) | 0.05-0.12 | 0.6555 | 0.11-0.14 | 1-1.01 | 7.25-8.77 | 91.26-94.82 | 4.59-6.9 | 0-0.04 | 0.57-1.84 | 0-0 | 287.1-945 | 4.86-8 | 3.75-8 | 3,4,5 | NA | NA |

*$c/\mu$ calculated with reference to Orf1ab 5'UTR mutation rate (lower boundary, $\mu$=36.1E-03% substitutions/nucleotide site/month; c=total substitutions/any site * year). The upper boundary of $\mu$ (i.e., determined from G15S of NSP5: $c/\mu$=2.42 in Fig 5) is 2.42-fold higher than the lower boundary of $\mu$ ($c/\mu$=1). The values of each parameter ($f_o$, WNB, NB, WPB, $f$, $f_o$, $f_o^+$, $f^+$, $f_o':f_o$, $f_o:f_o^+$, $f_o':f_o^+ + f^+$) for the lower and upper boundaries of $\mu$ are shown as ranges. Selection type is beneficial (+) if $c/\mu>1$, negative (−) if $c/\mu<1$ or near-neutral if $c/\mu\approx1$. $c/\mu$ is classified with letters for high $R^2$ (H, $R^2>0.6000$), low $R^2$ (L, $R^2<0.6000$), beneficial $c/\mu$ (+) or negative $c/\mu$ (−) values. See Table of S1 and S2 Tables for the average absolute substitution rate (c) value for the genome and each segment calculated from the three datasets.

**The fractions of sites under near neutral selection ($f_o'$) is equivalent to $c/\mu$; the fraction of sites under strict neutral selection ($f_o$) is obtained from $c/\mu$ discrete *DFE*; the fraction of sites under strong negative selection ($f$(SN)), weak negative selection ($f_o$(WN)), weak beneficial selection ($f_o^+$ (WP)) and strong beneficial selection ($f^+$ (SP)) are obtained from the cumulative *DFE*.

***WNB**: Weak negative selection boundary; **NB**: Neutral selection boundary; **WPB**: Weak beneficial selection boundary.

****Violation descriptors for *KNT*, *ONNT* and *ST*. Blank site**: No violation; **1**: %WP ≫ 0; **2**: %WN ≫ 0; **3**: %NN ≫ 0; **4**: %P(WP+SP) ≫0; **5**: %NN ≫ 0; **5**: fo ≫ fo. **NA**: Not applicable.

**Table 5. Relative rate and mutation fraction values for non-molecular clock SARS-CoV-2 genomic segments.** Position-based $c/\mu$, $R2$ and percent abundance at three selection type boundaries, the relative percent abundance of sites under five different selection types, proportion ratios and evaluation of $KNT$, $ONNT$ and $ST$ for each segment not exhibiting strict molecular clock of SARS-CoV-2 over 19 months, in order of descending average $c$ $R2$.

| Seg (NT length) | $f'_o = c/\mu$ | $c$ $R^2$ | WNB ($c/\mu$) | NB ($c/\mu$) | WPB ($c/\mu$) | $f$ (SN) | $f'_o$ (WN) | $f_o$ (N) | $f^+_o$ (WP) | $f^+$ (SP) | $f'_o{:}f_o$ (NN:N) | $f'_o{:}f^+_o$ (WN:WP) | $f'_o{:}f^+_o+f^+$ (WN:P) | KNT | ONNT | ST |
|---|---|---|---|---|---|---|---|---|---|---|---|---|---|---|---|---|
| Nsp5 (1,498) | 0.03-0.07 | 0.5965 | 0.14-0.14 | 1-1.01 | 7.02-7.25 | 92.81-96.29 | 3.26-5.99 | 0-1.2 | 0-0.43 | 0-0.03 | 240.98-1751 | 7.5-8.8 | 7.5-8 | NA | 1,4,5 | 2,3 |
| N 5'UTR (14) | 0.31-0.76 | 0.5577 | NA | 1.01-2.8 | NA | 64.28-100 | 0-35.71 | 0-0 | 0-0 | 0-0.05 | NA | NA | NA | NA | 5 | None |
| M 5'UTR (50) | 0.02-0.05 | 0.4925 | NA | 1.01-1.2 | NA | 98-100 | 0-2 | 0-0 | 0-0 | 0-0.06 | NA | NA | NA | NA | 1,5 | 2,3 |
| Orf8 5'UTR (134) | 0.09-0.23 | 0.4862 | 0.07-0.08 | 1.01-1.08 | 9.61-14.51 | 79.1-90.29 | 9.7-20.15 | 0-0.75 | 0-0 | 0-0.09 | NA | NA | NA | NA | 1,5 | 2,3 |
| Nsp14 (1,036) | 0.02-0.06 | 0.4841 | 0.12-0.17 | 1.01-1.45 | 5.8-8.26 | 94.12-97.1 | 2.4-5.2 | 0-0.67 | 0-0.48 | 0-0.01 | 465-871.92 | 5-18.15 | 5-18.15 | NA | 1,5 | 2,3 |
| Orf8 TRS-B (7) | 0.1-0.26 | 0.4698 | 0.34-0.49 | 0.82-1.01 | 2.9-10.74 | 71.43-85.71 | 14.28-14.29 | 0-14.29 | 0-0 | 0-0.62 | NA | NA | NA | NA | 1,5 | 2,3 |
| Orf7a (366) | 0.06-0.26 | 0.4031 | 0.07-0.12 | 1.01-1.21 | 8.26-14.51 | 73.77-93.16 | 6.28-25.41 | 0.01-0.82 | 0-0.54 | 0-0.01 | 550-1934.92 | 11.5-15.99 | 11.5-15.99 | NA | 1,5 | 2,3 |
| Orf10 (117) | 0.05-0.11 | 0.3557 | 0.17-0.17 | 1.01-1.04 | 5.61-5.8 | 88.89-93.16 | 5.98-8.55 | 0.02-2.56 | 0-0.85 | 0-0.11 | 102.96-320 | 4.67-7 | 4.67-7 | NA | 1,5 | 2,3 |
| Orf7a TRS-B (7) | 0.1-0.24 | 0.3129 | 0.28-0.42 | 0.82-1.01 | 3.63-10.29 | 71.43-85.71 | 14.28-14.29 | 0-14.29 | 0-0 | 0-1.1 | NA | NA | NA | NA | 1,5 | 2,3 |
| Nsp7 (868) | 0.02-0.05 | 0.3018 | 0.16-0.28 | 1.01-1.19 | 3.63-6.1 | 94.38-96.78 | 3.21-4.42 | 0-1.21 | 0-0 | 0-0.02 | NA | NA | NA | NA | 1,4,5 | 2,3 |
| Orf6 5'UTR (10) | 0.01-0.03 | 0.2871 | 0.02-0.24 | 1.01-1.01 | 1.51-4.15 | 80-90 | 20-Oct | 0-0 | 0-0 | 0-0 | NA | NA | NA | NA | 5 | 2,3 |
| Orf1ab TRS-L (7) | 0-0.01 | 0.2699 | NA | 1.01-1.01 | NA | 57.14-100 | 0-42.85 | 0-0 | 0-0 | 0-0 | NA | NA | NA | NA | 5 | 2,3 |
| Orf7a 5'UTR (6) | 0.12-0.29 | 0.2692 | 0.34-0.42 | 0.82-1.01 | 2.9-12.03 | 66.67-83.33 | 16.66-16.67 | 0-16.67 | 0-0 | 0-1.28 | NA | NA | NA | NA | 1,5 | 2,3 |
| N TRS-B (7) | 0.01-0.02 | 0.2535 | NA | 1.01-1.01 | NA | 85.71-100 | 0-14.28 | 0-0 | 0-0 | 0-0 | NA | NA | NA | NA | 5 | 2,3 |
| Orf10 5'UTR (24) | 0.12-0.3 | 0.2323 | 0.19-0.23 | 1.01-1.9 | 5.28-12.41 | 70.83-87.5 | 8.33-25 | 0.05-4.17 | 0.4-4.16 | 0-0.06 | 210-504 | 5-Feb | 5-Feb | NA | 1,5 | 2,3 |
| Orf10 3'UTR (229) | 0.15-0.37 | 0.1415 | 0.12-0.17 | 1.01-1.02 | 5.8-15.66 | 62.88-82.96 | 16.59-32.31 | 0-4.8 | 0-0.43 | 0-0.44 | 87.02-3705 | 1.9-38 | 1.73-38 | NA | 1,4,5 | 2,3 |
| E 5'UTR (24) | 0.01-0.03 | 0.0399 | NA | 1.01-1.01 | NA | 95.83-100 | 0-4.16 | 0-0 | 0-0 | 0-0 | NA | NA | NA | NA | 5 | 2,3 |
| S 5'UTR (7) | 0-0.01 | 0.0239 | NA | 1.01-1.01 | NA | 85.71-100 | 0-14.28 | 0-0 | 0-0 | 0-0 | NA | NA | NA | NA | 5 | 2,3 |
| Orf6 (186) | 0.03-0.07 | 0 | 0.07-0.12 | 1.01-1.16 | 8.26-14.51 | 92.43-95.16 | 4.3-5.96 | 0.01-1.61 | 0-0.53 | 0-0.03 | 260.4-342 | 7.67-8 | 7.67-8 | NA | 1,5 | 2,3 |
| S TRS-B (7) | 0-0 | -0.0075 | NA | 1.01-1.01 | NA | 85.71-100 | 0-14.28 | 0-0 | 0-0 | 0-0 | NA | NA | NA | NA | 5 | 2,3 |
| Orf3a 5'UTR (8) | 0-0 | -0.0235 | NA | 1.01-1.01 | NA | 87.5-100 | 0-12.5 | 0-0 | 0-0 | 0-0 | NA | NA | NA | NA | 5 | 2,3 |
| Orf3a TRS-B (7) | 0-0.01 | -0.0402 | NA | 1.01-1.01 | NA | 85.71-100 | 0-14.28 | 0-0 | 0-0 | 0-0 | NA | NA | NA | NA | 5 | 2,3 |
| Orf6 TRS-B (6) | 0.01-0.02 | -0.211 | NA | 1.01-1.01 | NA | 83.33-100 | 0-16.66 | 0-0 | 0-0 | 0-0 | NA | NA | NA | NA | 5 | 2,3 |
| E TRS-B (6) | 0-0 | NA | NA | 1.01-1.01 | NA | 100-100 | 0-0 | 0-0 | 0-0 | 0-0 | NA | NA | NA | NA | 5 | None |
| M TRS-B (7) | 0-0 | NA | NA | 1.01-1.01 | NA | 100-100 | 0-0 | 0-0 | 0-0 | 0-0 | NA | NA | NA | NA | 5 | None |

*See Table 1 for legend.

0.00% of 7 *TRSs* and 8 *UTRs* to 16.67% of Orf7a 5'UTR, 14.29% of Orf7a and Orf8 TRS-B, 0.85% of Orf10, 0.44% of Orf10 3'UTR, 0.40% of NSP7 and 0.05% of NSP5). From *KNT*, we would expect the overall fitness change including the contribution from nucleic acid fitness is due to the high $f_0$ for each genomic segment. However, this is not the case at least for SARS-CoV-2, suggesting the strict molecular clock feature for the 24 genomic segments does not arise through strictly neutral mutations, but perhaps mutations under natural selection or nearly neutral selection. The observed L-shaped distribution differs from previously reported *DFEs* that are highly-skewed distributions towards deleterious selection, which predict negligible fractions of nearly neutral mutations under weakly deleterious selection, weakly beneficial selection and strongly beneficial selection [30]. Walker et al. discuss the *DFEs* applying to both *TR* and *UTR* of a genome; in the *TR*, the *DFE* of nonsynonymous nucleotide sites is predicted by either 1) complex, multi-modal distributions (multiple peaks in *DFE*; a U-shaped *DFE* for *ST*, or an asymmetric Poisson-shaped *DFE* for *ONNT*) or 2) via a leptokurtic gamma distribution with a peak near strict neutral selection and an L-shaped tail towards strong deleterious selection. Conversely, in *UTR*, the *DFE* of non-coding nucleotide sites predicts peaks towards strict neutrality. In either case, the predictions by these *DFEs* cannot explain the non-negligible fractions of nucleotide mutations under weakly deleterious, weakly beneficial and strongly beneficial selection observed in this study.

### Varying computed fractions of nucleotide sites under different mutation types observed

The discrete $c/\mu$ *DFE* was transformed into the cumulative $c/\mu$ *DFE* to obtain the fractions of different mutation types within the lower and upper boundaries of $\mu$ ($c/\mu$ = 1 to 2.42, see Fig 3) (see Methods section for how the cumulative $c/\mu$ *DFE* was constructed and fractions of different mutation types were computed). The cumulative $c/\mu$ *DFE* of the SARS-CoV-2 genome indicate the nucleotide mutations under strong deleterious selection/$f^-$ (84.21 to 92.12%), slightly deleterious selection/$f_o^-$ (7.19 to 14.28%), weakly beneficial selection/$f_o^+$ (0.56 to 1.33%) and strongly beneficial selection/$f^+$ (0.10 to 0.19%) (see Tables 4 and 5, Fig 4 and Figures in S15 and S16 Figs). Of the substitutions that fix in the population (i.e., not strongly deleterious), those under slightly deleterious selection (90.38% to 91.59% (7.19 to 14.28%)), slightly beneficial selection (7.13% to 8.41% (0.56 to 1.33%)) and strongly beneficial selection (1.20% to 1.27% (0.10 to 0.19%)) are in order of increasing rarity.

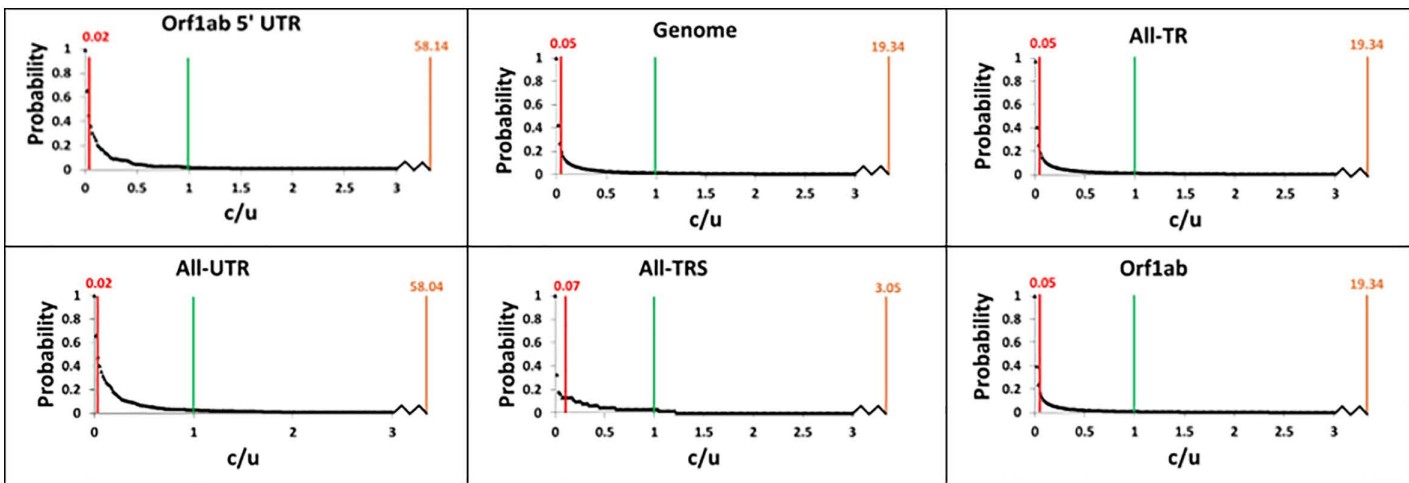

**Fig 3. Cumulative $c/\mu$ DFE of Orf1ab 5'UTR, genome, All-TR, All-UTR, All-TRS and Orf1ab.** The segments demonstrate a strict molecular clock, showing the abundance of sites under five different selection types ($f^-$, $f_o^-$, $f_o$, $f_o^+$ and $f^+$). The boundaries for weak negative selection (red line) to neutral selection (green line) to weak beneficial selection (orange line) and their determined $c/\mu$ positions are shown. Strong negative selection and strong beneficial selection would be to the left and right of the red and orange lines, respectively. A broken line leading on the x-axis represents the distance between $c/\mu$ = 3.0 to the weak beneficial selection boundary. See Figure in S11 Fig for the segments not demonstrating a strict molecular clock and Figure in S12 Fig for the remaining segments demonstrating strict molecular clock.

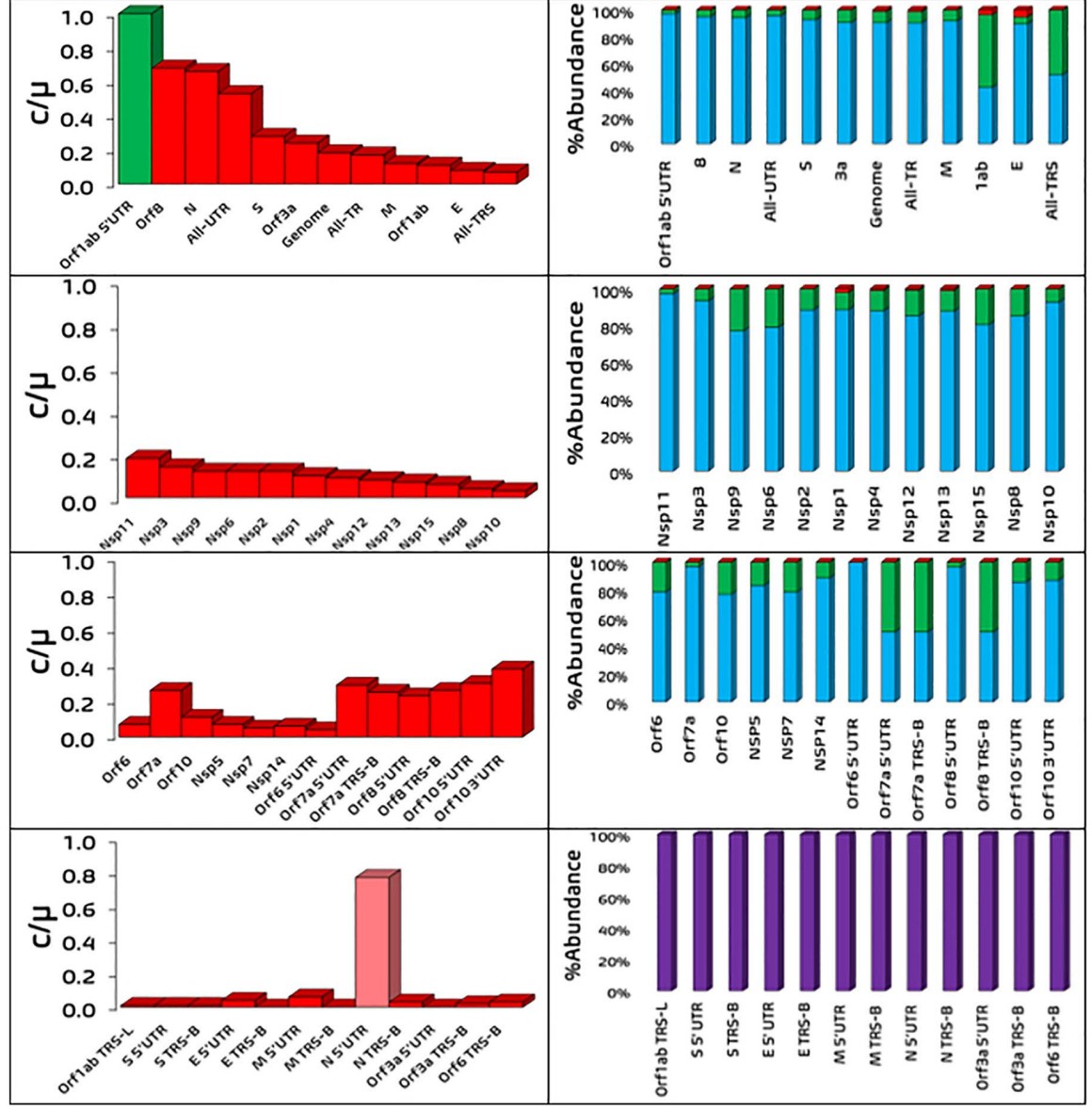

**Fig 4. Percent abundance of different mutation types within SARS-CoV-2 genomic segments.** *c/μ* values (left column) and abundance of selection type (right column) for segments exhibiting strict molecular clock features (first and second rows) and no strict molecular clock features (third and fourth rows). (Left column) Green, light red and dark red represent segments under neutral selection, weak negative selection and strong negative selection, respectively. (Right column) Blue, green, red and purple (where applicable) colors represent percentage of sites under weak negative, weak beneficial, strong beneficial and strong negative selection, respectively. See Figures in S15 and S16 Figs for the graphs of segments decomposed into other selection types. See Tables in S6 and S7 Tables for tabulated values.

## 2 to 9408-fold increase in $f'_0 : f_0$ for 34 genomic segments supports the importance of nearly neutral mutations over strictly neutral mutations contributing towards SARS-CoV-2 evolution

*KNT* and *ONNT* predict that the fraction of strictly neutral sites is much greater than the fraction of nearly neutral sites within a genomic segment ($f'_0 \ll f_0$). We thus calculated the $f'_0 : f_0$ for each SARS-CoV-2 genomic segment to determine

if this prediction holds true. We denote that if $f_0 = 0$ for a genomic segment, this will result in an $f_0' : f_0$ value that is undefined. $f_0' : f_0$ for the SARS-CoV-2 genome is~576 to ~1858. For the 24 genomic segments with strict molecular clock, their $f_0' : f_0$ ratio spans from ~4 to undefined of All-TRS to ~468 to ~3916 of Orf1ab, to ~587 to ~1600 of All-TR, and to ~817 to ~9408 of All-UTR (see Table 4 for remaining $f_0' : f_0$ values). For the 25 genomic segments without strict molecular clock, only 10/25 genomic segments exhibited a valid $f_0' : f_0$ ratio, ranging from ~2 to undefined of Orf7a 5'UTR to ~550 to ~1934.92 of Orf7a (see Table 5 for the remaining $f_0' : f_0$ values). From this, it seems the above prediction proposed by *KNT* and *ONNT* does not hold true for SARS-CoV-2; the higher fraction of nearly neutral mutation sites over strictly neutral mutation sites indicates the relative importance of the former towards the total substitution rate and their contribution towards generating the strict molecular clock feature of a genomic segment (i.e., $f_0' < f_0$).

### 1 to 31-fold increase in the $f_o^- : f_o^+$ ratio for 36 genomic segments supports the importance of weakly beneficial mutations contributing towards SARS-CoV-2 evolution

The revised prediction of *ONNT* proposes the fraction of weakly deleterious sites are greater than that of weakly beneficial sites within a genomic segment ($f_o^- \gg f_o^+$). We thus calculated the $f_o^- : f_o^+$ ratio to investigate this claim. We denote that if $f_o^+ = 0$ for a genomic segment, this will result in an $f_o^- : f_o^+$ value that is undefined. $f_o^- : f_o^+$ for the SARS-CoV-2 genome is~5 to ~12. For the 24 genomic segments with strict molecular clock, $f_o^- : f_o^+$ is applicable for 23/24 genomic segments, spanning from ~1 to undefined of NSP8 to ~31 of NSP9 (see Table 4 for the remaining $f_o^- : f_o^+$ values). For the 25 segments without strict molecular clock, $f_o^- : f_o^+$ is only available for 12/25 genomic segments, ranging from ~1–2 of Orf10 3'UTR to ~16 to ~4 to ~31 of Orf7a, to ~5 to ~18 of NSP14 (see Table 5 for the remaining $f_o^- : f_o^+$ values). It seems the revised prediction of *ONNT* does not hold for SARS-CoV-2; the non-negligible, sometimes greater fraction of weakly beneficial mutation sites over weakly deleterious mutation sites exists for some genomic segments (i.e., $f_o^+ > 0$ *and* $f_o^+ > f_o^-$) and the effects of these weakly beneficial mutation sites likely influence the balancing or misbalancing of the strict molecular clock feature.

### 1 to 37-fold increase in the $f_o^- : f_o^+ + f^+$ ratio for 37 genomic segments supports the non-negligible and sometimes greater contribution of beneficial mutations over weakly deleterious mutations contributing towards SARS-CoV-2 evolution

*ST* predicts the fraction of strongly beneficial sites is significant ($f^+ \gg 0$) and in greater abundance than the fractions of nearly neutral and strictly neutral sites within a genomic segment ($f^+ \gg f_o$ *and* $f^+ \gg f_o$), which is refuted by both *KNT* and *ONNT*. Predictably, we calculated the $f_o^- : f_o^+ + f^+$ ratio for each SARS-CoV-2 genomic segment to evaluate this prediction. $f_o^- : f_o^+ + f^+$ of the genome is~9 to ~11. For the 24 genomic segments with strict molecular clock, $f_o^- : f_o^+ + f^+$ spans from 0.73 to 13.18 of Orf1ab to 36.33 to 37.34 of NSP11 (see Table 4). For the 25 genomic segments without strict molecular clock, $f_o^- : f_o^+ + f^+$ is only available for 12/25 genomic segments, ranging from 1.73 to 38 of Orf10 3'UTR (see Table 5). The two ratios ($f_o^- : f_o^+ + f^+$) and ($f_o^- : f_o^+$) are nearly equivalent to each other because $f^+$ is very small for many sites (see Tables 4 and 5). Therefore, *ST* seemingly overestimates $f^+$ for SARS-CoV-2 due to its rarity (i.e., $0.00\% < f^+ \leq 1.28\%$), however $f^+$ for most genomic segments is not zero for this virus.

### While *NNST* is consistent with 6 empirical features of 49 SARS-CoV-2 genomic segments, *KNT*, *ONNT* and *ST* have many violations against these features

The summary of the theory evaluation for all SARS-CoV-2 genomic segments is tabulated in Table 6 and the violation descriptors for *ST*, *KNT* and *ONNT* for each genomic segment is available in Tables 4 and 5 (see Methods for more detail on criteria definition). For clarity, we first discuss the theory evaluations against genomic segments exhibiting the strict molecular clock (*KNT* evaluation), then against genomic segments not exhibiting the strict molecular clock (*ONNT* and *ST* evaluation). For the 24 genomic segments with strict molecular clock, *KNT* is not consistent with criteria 3, 4 and 5

**Table 6. Evaluation of *KNT*, *ONNT*, *ST*, *NNBST* and *NNUST* for all SARS-CoV-2 segments.** See Tables 2 and 3 for the evaluation at each segment. The number and percentage of violations calculated for applicable segments are presented.

| Theory | NA | APP | 1 | 2 | 3 | 4 | 5 |
|---|---|---|---|---|---|---|---|
| *KNT* | 25 | 24 | 0 | 0 | 24 (100%) | 16 (66%) | 24 (100%) |
| *ONNT* | 24 | 25 | 13 (52%) | 0 | 0 | 4 (16%) | 15 (60%) |
| ST | 24 | 25 | 0 | 22 (88%) | 22 (88%) | 0 | 0 |
| *NNBST* | 25 | 24 | 0 | 0 | 0 | 0 | 0 |
| *NNUST* | 24 | 25 | 0 | 0 | 0 | 0 | 0 |

\*NA: Not applicable to explain molecular clock; APP: Applicable to explain molecular clock.

*KNT*, *NNBST*: Applicable if $R^2 > 0.6000$; *ONNT*, *ST*, *NNUST*: Applicable if $R^2 < 0.6000$ (L+).

in 100%, 66%, and 100% of cases, respectively (see Tables 4 and 5). In other words, *KNT* overestimates the fraction of strictly neutral sites (criteria 5), underestimates the fractions of near neutral sites (criteria 3) and beneficial sites (criteria 4). For the 25 genomic segments without strict molecular clock, *ONNT* is not consistent with criteria 1, 4 and 5 in 52%, 16% and 60% of cases, respectively (see Tables 4 and 5); *ONNT* overestimates the fraction of strictly neutral sites (criteria 5), underestimates the fractions of weak beneficial sites (criteria 1) and beneficial sites (criteria 4). For the 25 genomic segments without strict molecular clock, *ST* is not consistent with criteria 2 and 3 in 88% and 88% of cases, respectively (see Tables 4 and 5); *ST* underestimates the fractions of weak negative sites (criteria 2) and near neutral sites (criteria 3). While *NNBST* is consistent with the strict molecular clock seen in 24 genomic segments (see Table 4), *NNUST* is consistent with the absence of the strict molecular clock seen in 25 genomic segments (see Table 5). Without the strong assumptions in the three conventional theories, *NNBST* and *NNUST* have no violations on criteria 1–5, indicating the 49 genomic segments of SARS-CoV-2 more likely follow *NNBST* and *NNUST*.

### NNST remains better over KNT, ONNT and ST to explain the data when switching to more conservative Ohta's boundaries, leading to a greater prevalence of adaptive substitutions, especially in *UTR* regions

To see if our theory evaluation conclusions can survive under varying *DFE* boundaries, we compare the fractions of different mutation types measured between our approach (using empirical $c/\mu$ data and symmetry) and Ohta's conventional approach (using an arbitrary cutoff and symmetry), which differ in defining the *WNB* and *WPB*. We use the SARS-CoV-2 genome, All-TR and All-UTR as examples (see Table in S8 Table). In detail, Ohta prescribed an arbitrary upper boundary for mutations under weakly beneficial selection (*WPB* $c/\mu = 2$) and a lower boundary for mutations under weakly deleterious selection assuming symmetry (*WNB* $c/\mu = 0.5$) [8,31], with a strictly neutral mutation boundary (*NB* $c/\mu = 1.0$) separating the two mutation types. The $c/\mu$ discrete *DFE* and $c/\mu$ cumulative *DFE* for the genome, All-TR and All-UTR revealed the fractions of strongly deleterious mutations/$f^-$ (Ohta: 93.51–97.2%; Our approach: 33.98–85.11%), weakly deleterious mutations/$f_o^-$ (Ohta: 1.22–3.63%; Our approach: 13.44–63.17%), strictly neutral mutations/$f_0$ (Ohta: 0.03–0.07%; Our approach: 0.03–0.07%), weakly beneficial mutations/$f_o^+$ (Ohta: 0.75–1.69%; Our approach: 1.25–2.85%) and strongly beneficial mutations/$f^+$ (Ohta: 0.74–1.17%; Our approach: 0.00–0.20%), including the normalized fractions of mutations that fix in the population for weakly deleterious mutations/$f_o^-$ (Ohta: 45.86–55.93%; Our approach: 90.38–95.68%), weakly beneficial mutations/$f_o^+$ (Ohta: 26.04–26.50%; Our approach: 4.32–8.42%) and strongly beneficial mutations/$f^+$ (Ohta: 18.03–27.82%; Our approach: 0.00–1.34%). The mutation fraction ratios were also computed for $f_0 : f_0$ (Ohta: 61.50–68.44; Our approach: 576.70–715.10), $f_o^- : f_o^+$ (Ohta: 0.84–1.28; Our approach: 2.15–5.54) and $f_o^- : f_o^+ + f^+$ (Ohta: 1.74–2.15; Our approach: 9.31–22.14) (see Table of S8 Table). The absolute fractions increased by ~11- to ~17-fold for weakly deleterious mutations/$f_o^-$ and ~1.4- to ~1.6-fold for weakly beneficial mutations/$f_o^+$, but decreased to ~1- and ~0.17-fold for strongly beneficial mutations/$f^+$ using our approach compared to Ohta's approach.

From these results, the percentage of strongly deleterious and strongly beneficial mutations decreased and the percentage of nearly neutral mutations (both weakly deleterious and weakly beneficial) increased when using our method, compared with Ohta's method. Consequently, the fraction ratios $f_0' : f_0$, $f_o^- : f_o^+$ and $f_o^- :  f_o^+ + f^+$ all decreased in Ohta's method because the fractions of nearly neutral mutations/$f_0'$ had decreased, since the *WNB* and *WPB* range is narrower in Ohta's method (*WNB* $c/\mu$ 0.52 and *WPB* $c/\mu = 2.02$ for the genome) compared to the much broader *WNB* and *WPB* ranges in our method (*WNB* $c/\mu = 0.07$ to 0.18 and *WPB* $c/\mu = 14.04$–19.34 for the genome). Our method captures a greater fraction of nearly neutral mutations/$f_0'$ compared with Ohta's method. Moreover, the absolute and normalized mutation fractions and their ratios suggest that weakly beneficial and strongly beneficial mutations are not rare in either our method or Ohta's method: the fraction of strictly neutral mutations are overestimated compared to nearly neutral mutations ($f_0' \gg f_0$); the fraction of weakly beneficial mutations are not negligible nor rare ($f_o^+ > 0$) and surprisingly close in abundance to the fraction weakly deleterious mutations ($f_o^- >  f_o^+$); the fraction of weakly deleterious mutations do not grossly outweigh the fractions of beneficial mutations ($f_o^- > f^+$, but $f^+ > 0$). (see Table of S8 Table). These results are inconsistent with *ONNT* but are consistent with *NNST*. Nonetheless, the non-negligible computed fractions of nearly neutral mutations in ours and Ohta's methods are inconsistent with the prediction of highly-skewed *DFEs* toward deleterious selection that these mutations are very rare.

## Discussion

If the three key assumptions regarding our replication selection model to describe the fixation of a mutation in the species' effective population are satisfied (see Methods section), the classical population-genetics equation relating the substitution rate ($c$) to the mutation rate ($\mu$), and an effective population size ($N_e$) to the fixation probability ($P_{fix}$) at a nucleotide site can be used to describe the evolution dynamics driven by the combined effects of mutation, genetic drift, and natural selection in a haploid population ($c = N_e * \mu * Pfix$). If mutations arise independently of $P_{fix}$, and $\mu$ at each site remains constant over the short time period (see Figure A of S1 Fig), using the site-specific $c/\mu$ test on empirical genomic sequence data can help resolve the "neutralist-selectionist" debate. By accurately quantifying the fractions of the five different mutation types ($f^-$, $f_o'$, $f_o$,  $f_o^+$ and $f^+$) at the nucleotide resolution for any genomic segment in *TR* and *UTR*/*TRS*, the $c/\mu$ framework generalizes the conventional *Ka*/*Ks* framework by providing the overall fitness change due to nucleic acid and protein fitness change, together, as opposed to just providing the protein fitness change alone and evaluates a genomic segment's molecular evolution against three conventional and two novel molecular evolution theories. Given the mutation profile's uniformity and the lack of high-resolution site-specific data to derive a site-specific mutation rate, as per our comparison between the observed substitutions from this study and cell-based and cell-free experimental studies (see Fig 1), we approximate SARS-CoV-2's mutation rate as a constant across all sites so that the site-specific $c/\mu$ analysis will directly tell us which molecular evolution theory it likely follows.

Several points can be made from our $c/\mu$ analysis for SARS-CoV-2. **First,** the L-shaped discrete $c/\mu$ *DFE* of its genome and its *TRs*, *UTRs* and *TRSs* differs significantly to the spike-shaped *DFE* of $c/\mu$ centered at 1 ($f_0 = 1$) under *GSRM*, the strictest form of *KNT* serving as a null hypothesis [17], the sharp Poisson-shaped $c/\mu$ *DFE* predicted by *KNT*, the Poisson-shaped $c/\mu$ *DFE* with asymmetric distribution of weakly deleterious mutations predicted by *ONNT* and the U-shaped $c/\mu$ *DFE* predicted by *ST*. This comparison suggests 1) the overestimation of genetic drifting (proposed by *ST*) and the underestimation of beneficial and deleterious selection (proposed by *KNT* and *ONNT*) in the fixing of mutations for SARS-CoV-2, and 2) the predominance of balanced nearly neutral mutations over strictly neutral mutations (proposed by *NNBST*) or unbalanced nearly neutral mutations (proposed by *NNUST*). **Second,** from the small fractions of strictly neutral mutations within each genomic segment, the genomic segments exhibiting a strict molecular clock (see Table 4) are likely due to nearly neutral mutations under a balanced condition mechanism (proposed by *NNBST*) rather than strictly neutral mutations under a genetic drifting mechanism (proposed by *KNT*). For the genomic segments not exhibiting a strict molecular clock (see Table 5), the contribution of weakly beneficial selection on the substitution rate is more consistent

with *NNUST* as opposed to *ONNT*. **Third,** *ONNT* predicts a population-size dependent substitution rate, since the lower purification efficiency acting on slightly deleterious mutations in a smaller population would lead to their higher substitution rate [14]. Yet, as the population size of SARS-CoV-2 varied over seven orders of magnitude ($10^0$–$10^7$) within the 19-month timespan of our genomic datasets, its computed *GSR* is relatively constant [17]. If the lower fixation efficiency on slightly beneficial mutations in the small population would lead to a lower substitution rate due to the increasing effects of genetic drift, then the higher rate from the slightly deleterious mutations will be cancelled out, and thus the population-size independent molecular clock can be observed. This mechanism is allowed in *NNBST*. However, asymmetric contributions of weakly deleterious and weakly beneficial selection likely abolish the strict molecular clock and produce a time-dependent substitution rate, allowing the unbalanced mechanism as per *NNUST*. **Fourth,** while non-coding nucleic acid sequences are assumed to undergo strict neutral selection, as proposed by *KNT*, the strict molecular clock feature was only observed for a few such segments (Orf1ab 5'UTR, All-UTR, All-TRS), indicating that stronger deleterious selection influences the time-dependent timelines of the remaining *UTRs* and *TRSs* for SARS-CoV-2. These findings are largely inconsistent with the neutrality assumptions by *KNT* but are more consistent with *NNBST* and *NNUST* (more so *NNUST*). **Fifth,** the abundance of nearly neutral substitutions, including weakly deleterious and weakly beneficial substitutions, are asymmetric **and** non-negligible, which differs from the asymmetric and negligible weakly beneficial substitutions (Ohta) and symmetric weakly deleterious and weakly beneficial substitutions (Gillespie) (see Table 2). Whether or not these fractions of nearly neutral substitutions are indeed equivalent to their stationary distributions (Sella and Hirsh) depends on if SARS-CoV-2 genome has also reached the stationary state (see Table 2), which is beyond the scope of this paper, but is under future investigation. **Sixth**, Ohta's boundary cutoffs for the *WNB* and *WPB* are defined independently of the observed $c/\mu$ distribution and may overlook the full range of nearly neutral mutations that drive evolutionary dynamics ($c = f'_o \mu_,$). In contrast, our approach defines the *WNB* and *WPB* based on segment-specific $c/\mu$ values, where the fraction of nearly neutral sites informs the boundary (*WNB* and *WPB* = 1/*WNB*). Applied to the SARS-CoV-2 genome, All-TR, and All-UTR, this method captures a broader and more representative spectrum of nearly neutral mutations than Ohta's definitions.

While most *TRs* in SARS-CoV-2 were classified as following *NNBST*, most *UTRs* and *TRSs* were classified as following *NNUST* based on the balanced/unbalanced forces of weakly beneficial and weakly deleterious selection, respectively. This assignment could reflect their distinct biological roles. *TRs* mostly preserve their encoded amino acid sequences and structural integrity to perform their critical function; most nonsynonymous nucleotide mutations which strongly impact fitness are not tolerated, but a limited number of weakly beneficial and weakly deleterious nucleotide mutations can modulate protein fitness, or nonsynonymous and synonymous nucleic acid fitness [20], offsetting each other's effects over evolutionary time. This could explain the strict molecular clock and coupled L-shaped $c/\mu$ *DFE* as per *NNBST*. In contrast to *TRs*, the roles of *UTRs* and *TRSs* are largely regulatory and functionally critical, and heavily depend on their sequence conservation. Along this line, the mutation effects of episodic weakly beneficial mutations that could alter RNA structures, transcription/translation efficiency or host factor interactions could be largely outweighed by the strongly deleterious mutation effects. This imbalance in the different mutation effects potentially cause non-strict molecular clock behavior as per the *NNUST*.

Additionally, we enhance upon several avenues on characterizing our $c/\mu$ and *NNST* frameworks first proposed in our previous paper [17]. **First**, we provide greater detail for our replication-selection model to obtain $\mu$, $c$ and $c/\mu$ and update our assumptions for the replication and selection processes. Our literature analysis suggests the polymerase copy error being the primary contributor towards $\mu$, which supports the general principle underlying the mutation-selection framework that a replication polymerase is blind to selection effects when making mutations. **Second**, we formalize *NNBST* and *NNUST* by providing a mathematical framework to explain the presence or absence of a strict molecular clock of a substitution timeline for a genome, *TR*, *UTR* or *TRS* under effective neutral selection; in our previous paper [17], we just described *NNBST* phenomenologically, but did not formally introduce *NNUST* at that time. Moreover, this framework challenges the proposal of *KNT*, suggesting that the strict molecular clock is generated by a balancing between the slower,

more abundant substitution rates under deleterious selection with the faster, less abundant substitution rates under beneficial selection. **Third**, we explicitly provide the genomic segment substitution timelines, discrete and cumulative $c/\mu$ *DFE*s and fractions of different mutation types for each *TR*, *UTR* and *TRS* for a single species for the first time, providing both a complete and personalized mutation spectra for a genome and its genomic segments. While it is tempting to only provide this information for the genome and "functionally important" genomic segments, such as druggable genes in *TR*, we avoid this bias here to support a standard for preliminary genomics analysis: each *TR*, *UTR* and *TRS* likely has functional importance, and their mutation spectra and mutation fractions should be provided when possible before more specialized analysis targeting specific genomic segments is performed. **Fourth,** we conducted a more rigorous evaluation of a genome and its genomic segments against the three conventional molecular evolution theories (*ST*, *KNT* and *ONNT*) and our *NNBST* and *NNUST* theories, adding the fractions of different mutation types, in addition to the presence of the strict molecular clock. Before, we phenomenologically characterized the strict molecular clock and presence of the L-shaped $c/\mu$ *DFE* as the criteria to dismiss *KNT*, *ONNT* and *ST* from explaining the SARS-CoV-2 *GSR*, instead following *NNBST* [17]. However, the specific violations induced against *NNBST*, *NNUST*, *ST*, *KNT* and *ONNT* cannot be inferred from these criteria alone. Here, the mutation fractions from empirical genomic sequence data allow more rigorous, quantitative determination if most mutation types fixed in a population agree or disagree with each of these five theories.

Nevertheless, our $c/\mu$ framework has some potential limitations. Across evolution time, the fitness changes within species populations are influenced by single point substitutions as well as polymorphisms, insertion-deletion mutations, recombination, epistatic interactions between individual nucleotide sites and genomic segments. So far, this $c/\mu$ framework only considers the independent selection effects within each nucleotide site and genomic segment, and has not yet been adapted to handle instances of epistasis between different nucleotide sites or genomic segments. Moreover, this framework quantifies $c$ and $\mu$ through empirical single-point substitution data, and has not yet considered insertion-deletion or recombination mutation data or gene migration phenomena. The genome-wide $\mu$ constancy assumption may be too simplistic when applied to the genomes of higher organisms, since $\mu$ is considered to vary across different genomic contexts [32], therefore this assumption can be relaxed to incorporate gene-specific $\mu$ or intragenic $\mu$ to more fairly reflect these contextual effects. Additionally, while $\mu$ was approximated from the substitution rate of SARS-CoV-2 Orf1ab 5'UTR, $\mu$ could also be approximated from intergenic regions or four-fold degenerate sites (D4 sites, in which mutations here do not change the amino acid sequence in *TR*) in this virus, or in other organisms depending on their substitution rate values. We emphasize that $\mu$ may not be strictly defined from either intergenic *UTRs* or D4 sites only, but that it will likely depend on the organism and available genomic data. Lastly, our $c/\mu$ framework has so far been validated for a single RNA virus with a relatively simple replication mechanism; applying our $c/\mu$ framework to other RNA viruses, microbial prokaryotes/ eukaryotes, plants and animal species will help to determine if it is also applicable to higher organisms in understanding their molecular evolution.

Our interpretation of strict and non-strict molecular clocks in the framework of *NNBST* and *NNUST* should be viewed alongside known evolutionary processes shaping SARS-CoV-2 and other organisms. For example, immune-driven selection from antiviral drugs, vaccines and host antibody responses can promote increased substitution rates at both antigenic sites in *TR* and regulatory sites in *UTRs*/*TRSs* [28], resulting in novel adaptive mutations that can antagonize these selection pressures and produce variants of concern (VOCs) that worsen human disease severity. In our genomic dataset, the absence of a strict molecular clock in Orf6 and Orf7a could be partially explained by their functional roles in modulating the host immune system and responding to host immune selection pressures over the course of the COVID-19 pandemic [33,34]. Transmission bottlenecks that reduce the viruses' effective population size during host infection can promote random fluctuations in fixing nearly neutral mutations into the viruses' genome; as effective population size becomes smaller, the effect of genetic drifting becomes stronger on mutation fixation, where mutations can be introduced or lost more randomly under these conditions, and strengthen the strict molecular clock feature [6,35]. Recombination is observed as a feature of later SARS-CoV-2 Omicron variants (e.g., Deltacron, XBB) [36], and recombination can generate

mosaic genomes whose local substitution patterns at each segment could result in non-clock-like behavior because new lineage founder viruses could exhibit different substitution patterns than the ancestor strain. In this study, the non-clock-like behavior of most *UTRs*/*TRSs* and some genes in *TRs* could arise from some of these biological processes and contribute to our motivation of introducing *NNUST*. Adaptive mutations in key *TRs,* such as the SARS-CoV-2 spike protein, were highly linked to evading the human immune response [28] and may also be linked to the increase in its substitution rate after the introduction of the vaccine [17]. The impact of recombination on our substitution timelines is likely minimal since our sequence datasets only contain pre-Omicron lineages. Furthermore, genomic segments which follow *NNBST* for this observed 19 month timespan could follow *NNUST*, or vice versa, if we consider, but not limit ourselves to the following scenarios: 1) We bias the geographical sampling of the SARS-CoV-2 genomes; 2) we change the genomic collection date timespan; 3) we change the reference sequence from the ancestor to more clade-specific reference sequences; 4) we introduce insertion/deletions, recombination data into the substitution rate calculations. Future work incorporating these changing processes will help refine and test this interpretation.

Lastly, while the present analysis has only been performed on SARS-CoV-2 genomic datasets, we emphasize that our $c/\mu$ framework can be extended beyond virus genomes; our interpretation is dependent on organism-specific genomic architecture, mutation processes and population genetic parameters. RNA viruses, including SARS-CoV-2, are characterized by compact RNA genomes, elevated mutation rates, and large effective population sizes [35,37,38]. Therefore, the assumption of a global, homogeneous genomic mutation rate ($\mu$) is a reasonable first-order approximation for RNA viruses. Conversely, DNA viruses and more complex organisms, including bacteria, plants and mammals, can exhibit heterogeneous mutation rates across different genomic regions via differences in replication timing, genomic architecture (e.g., chromatin structure), DNA repair and genetic linkage effects [32]. These conditions could cause local variations in $\mu$ and influence the inferred selection effects from $c/\mu$ at each nucleotide site and genomic segment. This does not invalidate our $c/\mu$ framework or *NNST*, but nonetheless necessitates the need for refined, context-dependent $\mu$ estimates when applied to more complex organisms. Future work will focus on estimating and incorporating region-specific $\mu$ estimates and population genetics parameters into our $c/\mu$ framework to systematically evaluate whether *NNST* is consistent for explaining the molecular evolution of numerous organisms with diverse genomic architectures and evolutionary dynamics.

phyloP is a well-known comparative-genomics method that quantifies how fast or slow each nucleotide site in a genome evolves relative to a neutral expectation [39–42]. Specifically, a site-specific rate-scaling factor ($r$), which measures how observed substitutions at a locus deviate from neutral expectations, is inferred from a multiple-sequence alignment together with a neutral evolutionary model estimated from putatively neutral sites (e.g., intergenic *UTR* regions). The phyloP score is reported as $-ln(r)$, such that positive values indicate conservation ($r<1$) and negative values indicate acceleration ($r>1$). In our framework, we define the $c/\mu$ ratio using the same rate-ratio structure, with $c$ representing the empirically observed substitution rate and $\mu$ representing the underlying neutral mutation rate. Accordingly, the $ln(c/\mu)$ selection score and the phyloP score differ only by a sign convention, establishing a direct mathematical equivalence between the two quantities. The key distinction lies in interpretation rather than form: phyloP derives $r$ empirically from cross-species alignments within a comparative-genomics framework, whereas $c/\mu$ is grounded in a mutation–selection framework informed by population-genetics theory, explicitly separating mutational input from selection-filtered evolutionary output. Despite these conceptual differences, both approaches quantify the same fundamental deviation from neutrality encoded in the rate ratio $c/\mu.$

phyloP has been extensively used as a genome-wide comparative-genomics method to quantify evolutionary conservation and acceleration across large multispecies alignments. The foundational work by Pollard et al. [39] introduced phyloP and analyzed 46-way vertebrate, 46-way placental mammal, and primate alignments, establishing the characteristic genome-wide distributions of conservation scores now widely used in human genome annotation (hg19). Subsequent studies expanded this framework to ~100 vertebrate species and, more recently, to ~240 mammalian genomes through large consortium efforts, greatly increasing power to detect weak constraint and lineage-specific acceleration in both coding and noncoding regions. Together, these studies demonstrate that phyloP robustly captures genome-wide deviations

from neutrality across dozens to hundreds of species, providing a mature empirical reference against which mutation–selection formulations such as $c/\mu$ can be directly interpreted.

Although genome-wide phyloP score distributions have been extensively generated and cataloged across many taxa, their implications for evolutionary theory have not been systematically examined. In most prior works, phyloP has been used primarily as an annotation or detection tool—to label conserved or accelerated sites—rather than as a quantitative object for testing population-genetics theories of molecular evolution. In this study, our framework explicitly bridges this gap by interpreting the phyloP distribution in evaluating evolution theory through the $c/\mu$ or $ln(c/\mu)$ distribution framework, which is directly grounded in mutation–selection theory. Because the phyloP score $-ln(r)$ is mathematically equivalent to $-ln(c/\mu)$ up to sign convention, the rich phyloP distributions already obtained from large multispecies alignments can be reinterpreted as empirical realizations of the $c/\mu$ distribution over long evolutionary timescales. This reframing enables, for the first time, a direct evaluation of whether the Near-Neutral Selection Theory (*NNST*)—originally developed in a population-genetics context of SARS-CoV-2—also governs evolutionary dynamics at deep phylogenetic scales. Thus, rather than generating new comparative statistics, our framework leverages existing phyloP results to test evolutionary theory itself, transforming descriptive conservation maps into quantitative evidence for or against *NNST* across extended evolutionary history.

Before interpreting these phyloP distributions in the context of *NNST, KNT,* or *ST*, it is important to clarify why the discussion here focuses on the classic 46-species (vertebrate, placental mammal, and primate) phyloP analyses rather than more recent studies with substantially larger species counts. As argued above, our theoretical classification relies on the probability density function (*PDF*) of the log rate ratio—$ln(c/\mu)$ or its phyloP-equivalent—because the *shape* of this distribution (near-neutral mode, skew, and tail behavior) is what discriminates among competing evolutionary theories. The 46-way phyloP studies directly reported genome-wide *PDF*-style histograms, making them immediately suitable for this purpose. By contrast, many recent large-scale comparative efforts report only cumulative distribution functions (*CDFs*), summary statistics, or thresholded conservation tracks; the underlying *PDFs* are not explicitly available. Recovering *PDFs* from *CDFs* requires numerical differentiation and careful smoothing, which we identify as a near-term to-do task for extending the present *NNST*-based analysis to larger and more recent multispecies datasets.

Across primates, placental mammals, and all 46 vertebrates, the phyloP histograms (see Figure of S17 Fig) share a common genome-wide structure that is most consistent with Near-Neutral Selection Theory (*NNST*) rather than strict Kimura Neutral Theory (*KNT*) or strong Selectionist Theory (*ST*). All three distributions show a dominant peak near phyloP $\approx 0$ ($-ln\ r \approx 0$), ($r \approx 1$)), indicating that most nucleotides evolve close to neutrality, a central prediction of *NNST* that remains robust even at deep evolutionary scales. At the same time, the distributions are strongly asymmetric, with a heavy positive tail (conservation; ($r < 1$)) and a persistent negative tail (acceleration; ($r > 1$)). Under *NNST*, this pattern arises naturally from the continuous introduction of mutations combined with differential filtering by selection: slightly deleterious mutations inflate the conserved tail, while a smaller but nontrivial fraction of sites experiences acceleration due to relaxation of constraint or episodic shifts. In contrast, *KNT* does not predict a stable, genome-wide conserved skew without imposing ad hoc site classes, and strong *ST* would erode the near-neutral peak, which remains the most prominent feature in all three datasets. Quantitatively, the hg19 46-way tracks—each containing 2.85 billion nucleotide-level values—exhibit small but consistently positive means, increasing variance with evolutionary depth, deep negative minima, and expanding positive maxima, all pointing to a robust near-neutral core plus asymmetric tails.

A key additional insight comes from how distributional shape changes with evolutionary scale. The primate phyloP distribution shows an L-shaped profile, closely mimicking the $ln(c/\mu)$ distributions observed in rapidly evolving systems such as SARS-CoV-2, where recent divergence highlights an excess of strongly constrained sites relative to neutrality. As the evolutionary scope broadens to placental mammals and then all vertebrates, the relative abundance of sites under very strong negative selection is reduced, consistent with the averaging of diverging selective regimes over longer timescales. Importantly, however, the distributions do not collapse to a symmetric peak centered at zero, as would be expected

under strict *KNT*. Instead, they retain a persistent positive bias toward conservation, indicating that near-neutrality, rather than pure neutrality, governs genome evolution even over deep phylogenetic time. This scale-dependent transition—from L-shaped at shallow divergence to broadened yet biased at deep divergence—is precisely the qualitative signature predicted by *NNST*.

## Conclusion

In this paper, we formulate the *NNST* mathematical framework including *NNBST* and *NNUST* based on empirical sequence analysis of SARS-CoV-2 via our *c/μ* framework to provide an explanation of the molecular evolution of this virus. This study emphasizes the use of empirical sequence data when available, as opposed to simulation data, to robustly support the *NNST* framework for at least explaining one case of intra-species genomic evolution. Our comprehensive *c/μ* analysis across each gene, *UTR* and *TRS* in the SARS-CoV-2 genome suggests its notable departures from traditional evolutionary theories. By extending our investigation to 26 *TRs*, 12 *UTRs* and 10 *TRSs* of this virus, we observed both L-shaped discrete and cumulative *c/μ DFE*s and the variable presence of strict molecular clocks, supporting both *NNBST* and *NNUST* instead of *ST*, *KNT* and *ONNT*. Our findings significantly challenge the existing paradigms of evolutionary biology by suggesting neither strictly neutral mutations nor solely beneficial mutations dominate the genomic landscape. Instead, we propose that nearly neutral mutations including both slightly deleterious and slightly beneficial selection, influenced by both genetic drift and natural selection, dominate the genomic landscape. This nuanced understanding suggests that the fixation of mutations is influenced by a balance of selective pressures rather than by simple stochastic events or direct selection alone. The discovery of L-shaped *c/μ DFE*s across all studied segments, coupled with the variable presence of strict molecular clocks, underscores the dynamic nature of viral evolution and the need for a revised theoretical framework. The *NNST* that we propose offers such a framework, combining elements of traditional neutral and selectionist theories to more accurately reflect the intra-species evolutionary processes observed in SARS-CoV-2, other RNA viruses and possibly higher-level organisms. Although a simple organism such as SARS-CoV-2 is often modeled with a uniform mutation rate, local features (e.g., CpG context or proximity to regulatory elements) in SARS-CoV-2, other RNA viruses and more complex organisms can cause site-specific variation [32,43]. To approximate these effects without per-nucleotide data, future work may require feature-specific rates to capture systematic differences. Nonetheless, the *c/μ* framework presented here marks a significant step towards resolving long-standing debates in evolutionary biology by generalizing the standard *Ka/Ks* method and sets the stage for future investigations that will further elucidate the intricate dance of mutation, selection, and drift in shaping the genomes of higher-level living organisms (see Table of S9 Table). We therefore urge the re-examination of intra-species molecular evolution using empirical sequence data, to not only critically assess these conventional theories of evolution, but to reinterpret our current understanding on how genetic drifting and natural selection contribute towards species evolution. Last not least, By establishing the mathematical equivalence between phyloP scores ($-\ln r$) and the mutation–selection metric $\ln(c/\mu)$, our *c/μ* framework reinterprets existing genome-wide phyloP distributions—previously used mainly for annotation—as theory-testing observables that enable direct evaluation of competing evolutionary models, including the Near-Neutral Selection Theory, across deep phylogenetic timescales well beyond SARS-CoV-2.

## Methods

For the ease of the reader, please refer to Table 3 for definitions of major symbols and terms mentioned throughout this paper.

### The *c/μ* method for quantifying the selection on all genome regions

In our recent paper, we developed a replication-selection model that uses site-specific and time-specific but site-independent and time-independent *c/μ* values to measure the transient selection coefficient acting on each nucleotide site in both *TR* and *UTR* of a virus genome using genomic sequence data for a short time period (see Figure A of S1 Fig) [17].

While our replication-selection model has been described in our recent papers [17,20], here we elaborate on our three basic assumptions made for the mutation rate ($\mu$) at the replication step in this model.

1) We assume that the introduction of mutations during replication period is independent of their probability of eventual fixation by selection, allowing $\mu$ and the fixation probability ($P_{fix}$) to factorize in determining the substitution rate (i.e., $c = N_e * \mu * P_{fix}$). Over a short time period ($t$), the probability of mutation introduction ($P_{mut}$) is theoretically quantifiable using in vitro cell-based or cell-free mutation accumulation or polymerase fidelity assays, where the genome replication enzymes such as polymerase, helicase, exonuclease etc., and the intrinsic local features of the nucleic acid template are likely the primary source of the mutation rate ($\mu = \frac{P_{mut}}{t}$) rather than by which proteins or nucleic acid elements will later be favored or disfavored. Because of the mutation–selection independence, the probability of a mutation's fixation in the population ($P_{sub}$) over the time period (i.e., $c = \frac{P_{sub}}{t}$) is measured by multiplying new mutation introduction ($N_e * \mu$) by $P_{fix}$ under selection. This assumption requires weak-mutation strong selection regime (WMSS): **1)** The $\mu$ is relatively small in viruses (i.e., $10^{-3}$ to $10^{-8}$ mutations per nucleotide site per year), where mostly single nucleotide mutations are introduced in the genome on a short time scale. **2)** Fixation or removal of the single point mutation in a relatively large effective population across a longer time scale is relatively fast so that it should be completed in the selection period before the next single point mutation is introduced into the genome. Host-driven selection acts both on exposed/antigenic proteins and on specific DNA/RNA elements that affect recognition, replication, or packaging without feeding back to alter the spontaneous error frequency at which those mutations first appear. Put together, the selection acting on a mutation at a site is site-specific but site-independent (no epistasis), hence $P_{fix}$ at one site is unaffected by $P_{fix}$ at other sites [44]. Whole genome mutation rate experiments—from molecular-clock analyses of neutral substitutions [45,46], and Luria-Delbrück fluctuation tests to mutation-accumulation experiments under relaxed selection [47] and in vitro polymerase fidelity assays [48,49] —all appear to show that mutations arise stochastically and independently of their fitness effects and that small changes in polymerase structure may not significantly impact its fidelity rate, supporting the factorization of mutation and selection in substitution rate models.

2) The site-specific $\mu$ is a time-independent constant over a short time-period observed within the sequence data, producing a strict molecular clock under strict neutral selection [7] within the time-period. This assumption is justified by the stable and low $\mu$ seen in viruses ($10^{-3} - 10^{-8}$ mutations per nucleotide site per year) so that mutations occurring in disjointed time intervals or at distinct sites are statistically-independent. At the molecular level, few mutations in the replication enzymes and the genome template should not significantly impact its fidelity rate over the short time period, ensuring the time-independent constancy of $\mu$. It should be noted that $\mu$ could change over longer time scales or across different sequence lineages. The time-dependent $\mu$ can be used to accommodate these effects, but it is beyond the scope of this study. Multiple independent lines of time-series and phylogenetic studies show that, over short observational windows, mutations accumulate at a steady, clock-like pace. (1) Longitudinal deep sequencing of viral populations sampled from individual hosts at successive time points reveals a linear increase in nucleotide changes per day, per week or per year [50,51]; (2) Controlled serial-passage experiments in cell culture or animal models document a constant number of new mutations per passage across dozens of transfers [38,52]; (3) Root-to-tip regressions of dated phylogenies display a strong linear relationship between sequence divergence and sampling time [53]; (4) Mutation-accumulation lines bottlenecked every generation accumulate fixed mutations in direct proportion to the number of generations sequenced [53,54] also for *C. elegans* [55] and yeast [56,57]. Together, these observations validate the assumption that, within the limited timeframe and sampling depth of most studies, the per-site $\mu$ can be treated as effectively constant.

3) We further assume that $\mu$ of simple haploid organisms like viruses and bacteria is about constant across all sites as the first order approximation due to the relatively uniform error profile of their replication enzymes and the lack of reliable,

high-resolution data on site-specific $\mu$ variation. Prior to a genome replication by a polymerase, helicase [58] uses energy from ATP to unwind double-stranded or folded DNA/RNA template, possibly removing contextual effects of the template sequence on the fidelity rate. The replicating polymerase can only interact with a small number of nucleotides in DNA/RNA template strand in the 5'◊3' elongation phase [59]; it probably has no sense of its global location in the template strand, be in the *TR* or the *UTR*. More specifically, the polymerase probably has no contextual sense of the genetic codon structure, including synonymous or nonsynonymous sites in *TR*, or ribosome-loading sites, transcription binding factors or promotor/regulation sites in *UTR*. These are justified from the order of the central dogma of molecular biology: the replication and transcription/translation processes occur separately and sequentially from each other, and the polymerase and ribosome do not interact with each other. Because of the very low $\mu$ in viruses ($10^{-3} - 10^{-8}$ mutations per nucleotide site per year), the polymerase will likely induce only ($1 - 10^{-5}$) nucleotide mutation per genome per replication cycle for a 1Kb virus genome. Clearly, the polymerase machinery is likely blind to the mutation's fate after committing a copy error (deleterious, neutral or beneficial) and consequently, it cannot preemptively adjust its $\mu$, nor adjust its bias in where it makes a mutation to account for the mutation's fitness effect. A wealth of studies across diverse systems shows that replication-error rates are effectively uniform across sites in simple haploids: mutation-accumulation experiments in yeast [57,60,61] and *C. elegans* [55,62] and HIV-1 [51]—where lines are bottlenecked each generation to minimize selection—reveal base substitutions scattered randomly rather than clustering at particular loci; ultra-deep sequencing of RNA virus populations (e.g., poliovirus) finds low frequency variants at synonymous and nonsynonymous sites occurring at comparable frequencies genome-wide [63,64]; in vitro fidelity assays of purified viral and bacterial polymerases on defined templates measure misincorporation rates that depend solely on enzyme kinetics and nucleotide chemistry (not much sequence context) [43]; and targeted reporter constructs inserted at different neutral chromosomal locations yield roughly equivalent reversion rates, indicating that local DNA context has little effect on the baseline $\mu$ [65,66]. Together, these lines of evidence validate the assumption that, for simple haploid organisms, the per-site $\mu$ might be treated as a constant across the genome.

This $c/\mu$ test is based on the well-established MutSel framework in population genetics (**Eq. 4** $c = N_e * \mu * P_{fix}$), where the nucleotide substitution rate ($c$) is determined by a product of the effective population size ($N_e$), the mutation rate ($\mu$), and the fixation probability of mutations in the effective population ($P_{fix}$). $P_{fix}$ of a mutation in a haploid population can also be described using a equation derived by Fisher [67], Wright [68], Kimura [69] and Halpern and Bruno [26]: (**Eq. 5a** and **5b** $P_{fix} = \frac{1-e^{-2s}}{1-e^{-2N_es}} \approx \frac{2s}{1-e^{-2N_es}}$), where $s$ is the transient selection coefficient of the mutation at the nucleotide site per individual; when $s$ is small ($|s|\ll1$), **Eq. 5a** is simplified to **Eq. 5b**. When $s$ is zero, $P_{fix}$ for a neutral mutation in a haploid population just depends on its initial frequency in the population, known as genetic drifting: (**Eq. 6** $P_{fix} = \frac{1}{N_e}$ *and* $c = u$). Otherwise, $c/\mu$ under selection can be obtained by substituting **Eq. 5** into **Eq. 4** ($c = N_e * u * P_{fix}$) and by setting a population scaled transient selection coefficient (**Eq. 7a** $S = 2N_es$ and **Eq. 7b** $\frac{c}{u} \approx \frac{S}{1-e^{-S}}$). Because $P_{fix}$ of neutral mutations is equal to $1/N_e$, $c/\mu$ becomes a measure of the selection acting on a specific nucleotide site within specific time (**Eq. 5b** $P_{fix} = \frac{1-e^{-2s}}{1-e^{-2N_es}}$). Thus, the $c/\mu$ ratio test measures the strength of selection pressure [70,71] and infers the total fitness effect at each nucleotide site in a genome, both in *TR* and *UTR,* over a given unit of time (strictly neutral if $c/\mu = 1$, deleterious if $c/\mu < 1$ and beneficial if $c/\mu > 1$). Because the fitness change of a single nucleotide site due to nucleotide mutation is independent from that of a different site (i.e., $s_i = s_1, s_2, \ldots, s_n$, where $n$ represents the total length of a genomic segment), the total fitness change for a gene, gene domain, *UTR* or genome can be easily obtained by first computing the average substitution-mutation rate ratio for a segment $\frac{\bar{c}}{\mu} = \frac{1}{\mu} \sum_{i=1}^{n} c_i$ and then use it to infer the average scaled selection coefficient for a segment by solving **Eq. 7b**. While calculating $s_i$ for each site can reveal the adaptive mutations, performing the same action for an entire gene segment can infer the different selection forces acting on these segments. For example, we expect the SARS-CoV-2 spike glycoprotein and nucleocapsid protein to be under stronger natural selection, moreso adaptive selection, due to their higher nucleotide and nonsynonymous mutation rates; in contrast, the nonstructural proteins are largely under stronger deleterious selection due to their roles in genome replication and immune suppression.

 

## Development of Near-Neutral Selectionist Theory (*NNST*) with a balanced condition (*NNBST*) or an unbalanced condition (*NNUST*)

Under the assumption that $\mu$ for different mutation types is the same as posed in our replication-selection model (see Figure A in S1 Fig), the contribution of each selection/mutation type towards the total substitution rate for a given genomic sequence, segment or collection of sites, can be described as the weighted sum of the substitution rates of sites under strong deleterious selection ($c^-f^-$), weak negative selection ($c_0^-f_0^-$), strictly neutral selection ($\mu f_0$), weak beneficial selection ($c_0^+f_0^+$) and strong beneficial selection ($c^+f^+$) by reducing *DFE* into five categories in **Eq. 8a**:

$$c = c^-f^- + c_0^-f_0^- + \mu f_0 + c_0^+f_0^+ + c^+f^+ \tag{8a}$$

Where the fraction of sites under each selection type (prefix *f*) must equal to one in **Eq. 8b**:

$$1 = f^- + f_0^- + f_0 + f_0^+ + f^+ \tag{8b}$$

**Eq. 8a** and **8b** can be reduced into **Eq. 9a** and **9b**, which follow *NNST* by assuming the substitution rate of strongly deleterious mutations and the fraction of strongly beneficial mutation is sufficiently small ($c^- \approx 0$ *and* $f^+ \approx 0$):

$$c \approx 0*f^- + c_0^-f_0^- + \mu f_0 + c_0^+f_0^+ + c^+0 \tag{9a}$$

$$1 \approx f^- + f_0^- + f_0 + f_0^+ + 0 \tag{9b}$$

Although the third neutral site term ($\mu f_0$) is time-independent due to $\mu$ being constant, the overall substitution rate is more likely to be time-dependent because of the time-dependence of the second weak beneficial site term ($c_0^+f_0^+$) and the fourth weak deleterious site term ($c_0^-f_0^-$). However, the total substitution rate can be time-independent under the balanced condition of *NNBST*, where the decrease of the rate due to the deleterious mutations ($c^-f^-$, $c^- < \mu$) is exactly canceled out by the increase of the rate due to the advantageous mutations ($c^+f^+$, $c^+ > \mu$) over the time period:

$$(\mu - c_0^-)f_0^- = (c_0^+ - \mu)f_0^+ \tag{10a}$$

**Eq. 10a** can be rearranged into **Eq. 10b**,

$$c_0^-f_0^- + c_0^+f_0^+ = \mu f_0^- + \mu f_0^+ \tag{10b}$$

where the left two time-dependent terms are effectively converted into the two-right time-independent terms. By substituting **Eq. 10b** into **Eq. 9a**, the total substitution rate can be rewritten into three terms with a common factor of $\mu$:

$$c = \mu f_0^- + \mu f_0 + \mu f_0^+ \tag{11c}$$

**Eq. 11c** can be simplified into **Eq. 12a and b**:

$$c = \mu f_0' \tag{12a}$$

$$f'_0 = f^-_0 + f_0 + f^+_0 = 1 - f^-,$$ (12b)

where $f'_0$ ($f'_0 \leq 1$) is defined as the equivalent fraction of nearly neutral mutations including the fraction of weak beneficial ($f^+_0$), strictly neutral ($f_0$) and weak deleterious ($f^-_0$)mutations (**12b**). Thus, the total substitution rate appears to be a time-independent rate constant due to the time-independence of $f'_0$. However, when the balanced condition is not satisfied,

$$(\mu - c^-_0)f^-_0 \neq (c^+_0 - \mu)f^+_0$$ (13a)

The total substitution rate will not be time-independent due to the time-dependence of $f'_0$:

$$c = c^-_0 f^-_0 + \mu f_0 + c^+_0 f^+_0$$ (13b)

Because the deleterious mutations are typically more abundant than beneficial mutations, it is more likely that the decreased substitution rate from slightly deleterious mutations is not balanced the increased substitution rate from slightly advantageous mutations over the measured time (**Eq. 13a**). **Eqs. 13a and 13b** define *NNUST*, predicting a time-dependent total substitution rate. In addition, the total substitution rate can be 1) lower than $\mu$ under effective deleterious selection ($f'_0 = c/\mu < 1$) or 2) be higher than $\mu$ under effective beneficial selection ($f'_0 = c/\mu > 1$) under rare cases.

**Comparison between *NNBST* to *KNT* and *ONNT* and comparison between *NNUST* to *ST***

*NNBST* can be reduced to *KNT*, which defines the overall substitution rate as the product of the mutation rate with the fraction of strictly neutral mutations only by assuming the fraction of beneficial mutations and the substitution rate of deleterious mutations are negligible ($f^+_0 \approx 0$ and $f^-_0 \approx 0$) and strictly neutral sites comprise the majority of the effective neutral mutations ($f'_0 \approx f_0$) [7]:

$$c = \mu f_0$$ (2a)

$$f_0 = 1 - f^-$$ (2b)

Conversely, **Eqs. 13a and 13b** of *NNUST* can be reduced to *ST*, which define the overall substitution rate would be the product of the substitution rate of strong beneficial sites ($c^+$) with its fraction ($f^+$) by assuming the fraction of nearly neutral mutations and the substitution rate of deleterious mutations are negligible ($f'_0 \approx 0$ and $c^- \approx 0$):

$$c = c^+ f^+$$ (1a)

$$f^+ = 1 - f^-$$ (1b)

In other words, only strongly beneficial mutations can be fixed in the population.

Several statements regarding *NNBST* and *NNUST* in comparison to *ST*, *KNT* and *ONNT* can be made.

**First**, regarding the fraction of nearly neutral sites, the balanced condition in *NNBST* allows a non-zero fraction of nearly neutral sites, opposite to *KNT*; the presence of a strict molecular clock is necessary but not sufficient to support *KNT* (i.e., $f_0 > f^+_0 + f^+$). Additionally, *NNUST* allows a greater non-zero fraction of nearly neutral sites than *ST* allows; conversely, the lack of a strict molecular clock is necessary but not sufficient to prove *ST* ($f^+ > f_0$).

**Second**, both *NNBST* (**Eq. 12a**) and *KNT* (**Eq. 2a**) predicts that a strict molecular clock with variable $f_0'$ or $f_0$ for different genome segments and the maximum substitution rate is limited by the mutation rate ($c \to u$ when $f_0'$ or $f_0 \to 1$) due to the impact of overall deleterious selection. In contrast, *NNUST* (**Eq. 13b**) and *ST* (**Eq. 1a**) predicts a time-dependent substitution rate due to time-dependent selection types: ($c_0^- f_0^- + c_0^+ f_0^+$) in *NNUST* and ($c^+ f^+$) in *ST*, leading to either an overall deleterious selection with a upper rate limit of μ or an overall beneficial selection without an upper rate limit.

**Third**, assuming a large fraction of nearly neutral mutations are slightly deleterious, *ONNT* predicts a deleterious correlation between the evolutionary rate and effective population size. As a result, the substitution rate becomes time-dependent due to changes in the effective population size over time, leading to the absence of a strict molecular clock. Another way to support this prediction is that since the balanced condition (**Eq. 12b**) is not satisfied due to the lack of slightly advantageous mutations ($f_0^+ \approx 0$), *ONNT* predicts the absence of a strict molecular clock (**Eq. 3**). Put together, *ONNT* is like *NNUST*.

**Fourth**, while *ONNT* and *KNT* suggest an abundance of strictly neutral mutations, *NNST* (*NNBST* and *NNUST*) does not.

**Fifth**, if the balanced condition is satisfied, then the strict molecular clock will be observed as predicted by *NNBST*. However, there is no good justification to support that beneficial selection must be balanced with deleterious selection, thus balanced selection should occur less frequently than unbalanced selection. In fact, the lack of a strict molecular clock was observed in 25 of 32 RNA viruses [72], including SARS-CoV-2 [17].

Key features of the expected *DFE* for different models/theories of molecular evolution are summarized in [Fig 5](). Assuming all nucleotide sites undergo strictly neutral selection only (i.e., $f_0 = 1$; Genomic Substitution Rate Model/*GSRM*; the strict neutrality null hypothesis [1,17]), the observed mutation rate at any nucleotide site $j$ (substitution rate, $c_j$) is equal to the spontaneous mutation rate in individual organisms ($c = \mu$) [3–5]. Empirical hallmarks of *GSRM* are 1) global molecular clock, 2) discrete almost uniform distribution of $c/\mu$ at any site of the genome, and 3) Poisson distribution centered at $c/\mu = 1$ with a broadened peak due to possible random sampling error. The Segment Substitution Rate Model/*SSRM* [17]), was introduced to allow the time-dependent natural selection pressure acting on the genome, thus causing the deviation of the substitution rate of a specific segment or a nucleotide site $j$ from the constant mutation rate (i.e., $\frac{c_j}{\mu} \neq 1$). The conventional theories including our *NNST* propose to capture the relative contribution between genetic drift and natural selection forces in determining the evolutionary substitution rate [1,17]). In *KNT*, the sites under strictly neutral selection are much more abundant than the sites under strongly beneficial selection ($f_0 \gg f^+$) [70], leading to the molecular clock feature and a Poisson distribution centered at $c/\mu = 1$. In *ONNT* [8,9], the sites under strictly neutral selection or nearly neutral selection are much more abundant than the sites under beneficial selection ($f^+ \approx 0$, $f_0' > f_0 \gg f^+$), leading to a time-dependent substitution rate [1] and an asymmetric distribution around $c/\mu = 1$ due to mostly slightly deleterious near neutral mutations ($f_0^- \gg f_0^+ \approx 0$). In *ST* [3], nucleotide positions largely exhibit natural selection ($\frac{c_j}{\mu} \neq 1$) and few sites are under strictly neutral selection, leading to a time-dependent substitution rate and a bimodal distribution of $c/\mu$ showing one peak of deleterious selection ($c/\mu < 1$) and the other peak of beneficial selection ($c/\mu > 1$). Conversely, in *NNBST*, the hallmarks are an L-shaped $c/\mu$ *DFE* and molecular clock due to the cancelling out of the rate of change of the sites under weakly beneficial selection by that of the sites under weakly deleterious selection. Although slightly deleterious mutations are more abundant than slightly advantageous mutations, both are not rare ($f_0^- > f_0^+ \gg 0$), whereas strictly neutral mutations are very rare ($f_0' \gg f_0$). Lastly, in *NNUST*, hallmarks are L-shaped distribution of $c/\mu$ and lack of molecular clock due to the failure of the balanced condition. Again, *NNUST* is more general than *ST* without assuming that nearly neutral mutations including slightly deleterious mutations are so rare.

## Computational workflow

In brief, our computational workflow utilizes $c/\mu$ analysis to accurately quantify the fractions of mutation types of a genome, including strictly neutral mutations ($c/\mu = 1$), contributing to timeline feature for a genomic segment. We followed

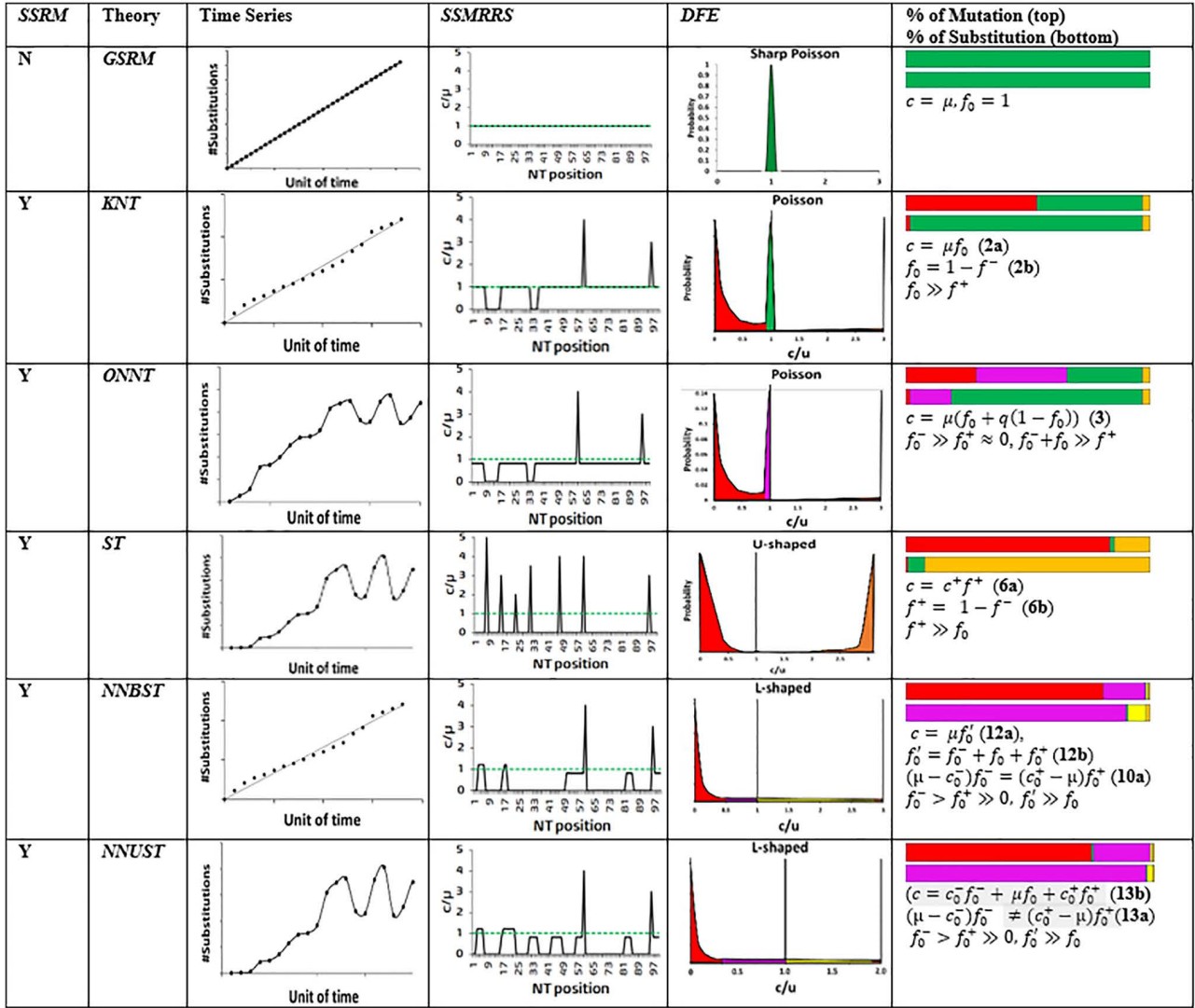

**Fig 5. Summary of the key features of molecular evolution theories.** Two substitution rate models (*GSR*m and *SSRM*) and five theories of molecular evolution (*ST, KNT, ONNT, NNBST, NNUST*) in terms of molecular clock, *SSMRSS*, *DFE* and percentage of sites under different selection types. *\*GSR*M (Genomic Substitution Rate Model/Strict neutrality hypothesis); *SSRM* (Genomic Substitution Rate Model); *ST* (Selectionist Theory); *KNT* (Kimura's Neutral Theory); *ONNT* (Ohta's Nearly Neutral Theory); *NNBST* (Near-Neutral Balanced Selectionist Theory); *NNUST* (Near-Neutral Unbalanced Selectionist Theory); *NNST* (Near-Neutral Selectionist Theory): including *NNBST* and *NNUST*; $\mathbf{c} = c^-f^- + c_0^-f_0^- + \mu f_0 + c_0^+f_0^+ + c^+f^+$ (**Eq. 24a**) and $1 = f^- + f_0^- + f_0 + f_0^+ + f^+$ (**Eq. 24b**) under *SSRM*: the mean substitution rate **c** for a given genomic sequence, segment or collection of sites, can be described as the weighted sum of the substitution rate of the sites under strong negative selection ($\mathbf{c}^-f^-$), weak negative selection ($\mathbf{c}_0^-f_0^-$), strictly neutral selection ($\mu f_0$), weak positive selection ($\mathbf{c}_0^+f_0^+$), and strong positive selection ($\mathbf{c}^+f^+$) by reducing the discrete $c/\mu$ distribution into these five categories; **c** : total substitution rate; $\mathbf{c}^-, c_0^-, \mu, c_0^+, and\ c^+$ : mean substitution rate of strongly deleterious, weakly deleterious, strictly neutral, weakly beneficial and strongly beneficial sites; $f^-, f_0^-, f_0, f_0^+$ and $f^+$: the fraction of strongly deleterious, weakly deleterious, strictly neutral, weakly beneficial and strongly beneficial sites; $\mathbf{f}_0' = f_0^- + f_0 + f_0^+$: the fraction of nearly neutral sites; $q$: time-dependent proportion of deleterious mutations that reach fixation; *\*\*Molecular clock presence (Yes/No); SSMRRS*: Site Frequency Spectra; *DFE*: Distribution of Fitness Effects; *WNB*: weak negative boundary; *WPB*: weak positive boundary; *SN*: Strong negative selection ($0 \le c/\mu < WNB$) in red color; *WN*: Weak negative selection ($WNB \le c/\mu < 1$) in pink; *N*: Selectively Neutral selection ($c/\mu = 1$) in dark green; *WP*: Weak positive selection ($1 < c/\mu \le WPB$) in light yellow; *SP*: Strong positive selection ($c/\mu > WPB$) in dark yellow.

a computational workflow (see Figure in S2 Fig for the raw data generation and analysis in our previous papers) [20,28]. The workflow proceeds in a 7-step process. **1 and 2)** Sequence compilation and MAFFT multiple sequence alignment of the three SARS-CoV-2 genomic sequence datasets (downloaded from NextStrain) comprising 11,198 total sequences from human patients was performed as in our previous protocol [17]. The Wuhan-Hu-1 SARS-CoV-2 genomic reference sequence was used for MAFFT alignment. **3)** Generate the nucleotide-based substitution matrix using the JC69 mutation model [73]. Calculate the total substitution rate ($c$) in both time-based and position-based methods as described in our prior study [20]. We detail the time-based approach later in the methods. **4)** The time-based calculations of ($c$) were used to generate the substitution timeline diagrams and coefficient of determination ($R^2$) to check for the molecular clock feature ($R^2 \geq 0.6000$) and to approximate $\mu$. The position-based calculations of ($c$) were used to generate the $c/\mu$ SSMRRS diagrams. **5)** Transform the $c/\mu$ SSMRRS into the discrete and cumulative $c/\mu$ DFE diagrams and infer the weak negative boundary ($WNB$) and weak positive boundary ($WPB$) for step 6. **6)** Calculate the proportion of strictly neutral mutations from the discrete $c/\mu$ DFE and the strongly deleterious, weakly deleterious, weak beneficial and strongly beneficial mutations from the cumulative $c/\mu$ DFE. **7)** Using both $R^2$ and mutation type proportions, evaluate each segment against the five theories of molecular evolution. Additional details for **steps 4–7** are outlined.

**Step 4A. Defining Segments to Measure Strict Molecular Clock Feature.** To search for genomic segments with molecular clock feature, a brute force strategy is to generate all possible segments of a genome of length $N$ nucleotide sites, calculate their substitution timeline, and determine linearity of the timeline ($c$ and $R^2$). **Eq. 14** counts the number of ways to form a non-empty segment from a set of $N$ nucleotide sites. Each term $\binom{N}{i}$ represents the number of ways to choose exactly $i$ sites from the $N$ available sites, where the order of selection does not matter. Adding these terms for $i$ from $1$ to $N$ gives the total number of ways to choose any number of sites, except for choosing none at all. Since every item can either be included or not included, there are $2^N$ possible combinations, but one of these combinations is the empty set. Therefore, when we subtract that one case, we end up with $2^N - 1$ as the total number of non-empty combinations.

$$S = 2^N - 1 = \sum_{i=1}^{N} \binom{N}{i}$$

(14)

For the SARS-CoV-2 genome with $N = 29,903$, $S = 2^{29,903} - 1$. One easily sees that $S$ is too large to be searchable. To reduce the search space, functional segments including virus structural and accessory genes (virus life cycle and host immune modulation) or $UTRs$ and $TRSs$ (ribosome loading and genetic regulation) can be used instead, because sites in these segments are naturally correlated (i.e., nucleotide site in one position is linked to other nucleotides in other positions within the same segment). In this study, our search was restricted to the 49 segments of SARS-CoV-2 (1 genome, 1 All-TR, 1 All-UTR, 1 All-TRS, 10 structural genes, 15 non-structural proteins, 11 $UTRs$, 9 $TRSs$) defined in our previous study [17].

**Step 4B. Statistical Determination of Constancy of Substitution Rate ($c$).** The empirical observation of strict molecular clocks is critical for determining the lower boundary of $\mu$ [20]. Linear regression is widely used in assessing the linearity of virus substitution timelines for determining constancy of substitution rate (i.e., the strict molecular clock feature). [4,37,72,74,75] Thus, the time-based approach with least squares fitting ($y = a * x$, $R^2$) was used to generate the segment substitution timeline to calculate the absolute monthly rate for SARS-CoV-2 from December 2019 to June 2021. By passing the timeline through the origin point of an X-Y Cartesian coordinate system, where x is the unit time (e.g., 0–19 months) and y is the percent total substitution rate per segment, the coefficient of determination ($R^2$) and slope ($c = \frac{y}{x}$) from the least square fit is obtained. $R^2$ measures the linearity of the percent substitution rate data and provides a statistical measure on the "strength" of the strict molecular clock feature if it is present, and thus provides the critical hallmark of effective neutral substitutions: a time-independent rate whose speed depends on an unknown fraction of effective neutral sites in the segment ($c = \mu f_0^e$). Since strict molecular clocks average out the signal noise in random

sequence sampling, the time-based and position-based substitution rate values ($c$) will differ by the scaling constant that is later removed from the final $c/\mu$ value once $\mu$ is approximated in Step 4. If the strict molecular clock is observed ($R^2 \geq 0.6000$), the time-based and position-based $c/\mu$ values can be similar to each other, and either the time-based or position-based $c/\mu$ values can be used to infer selection pressure. In cases where molecular clock is not observed ($R^2 < 0.6000$), $c/\mu$ values from the position-based approach should instead be used. While the strict molecular clock feature does not have to exhibit perfect linearity, our cutoff ($R^2 \geq 0.6000$) could be considered sufficiently relaxed, but not too stringent, for satisfying the strict molecular clock feature.

While $R^2$ was analyzed for each segment, the correlation coefficient ($r^2$) where the trendline does not pass through the origin of the Cartesian Coordinate system, could assess the strict molecular clock more accurately than $R^2$ if the sequence sample size is too small or multiple ancestor sequences are present in the evolution of the species. We therefore performed a pairwise comparison between $R^2$ and $r^2$ for the time-based *GSR* of SARS-CoV-2 in our previous paper, which showed insignificant differences between them ($R^2 = 0.9957$, $r^2 = 0.9714$) [20]. Herein, we perform only $R^2$ analysis for the genome and each segment in this study for simplicity.

**Step 4C. Choice of reference sequence on total substitution rate and molecular clock feature.** For SARS-CoV-2, several early pandemic genome sequences such as Wuhan-Hu-1 and Wuhan IPBCAMS-WH-01 can serve as the reference sequence for our substitution analysis. To quickly check if the *GSR* and strict molecular clock feature becomes significantly altered when different reference sequences are used, we generated *GSR* timelines using either Wuhan-Hu-1 or Wuhan IPBCAMS-WH-01 as the reference sequence and compared their absolute substitution rates and strict molecular clock assessment ($R^2 \geq 0.6000$). The absolute substitution rate changes slightly ($c = 6.4\text{E-}3$ to $7.2\text{E-}3$ per nucleotide per month) and the molecular clock shape is maintained ($R^2 = 0.9829$ to $0.9912$) when switching the reference sequence from Wuhan-Hu-1 to Wuhan IPBCAMS-WH-01 (see Figure in S5 Fig). Its effect on $c/\mu$ should be even smaller, thus Wuhan-Hu-1 was used as reference sequence in this study.

**Step 4D. Validation of the Fundamental Genomic Mutation Rate ($\mu$).** The fundamental genomic mutation rate ($\mu$) is a critical parameter of the $c/\mu$ framework. We approximated $\mu$ using the effective neutral substitution rate approach, which assumes that a subset of genomic sites are under near-neutral selection and evolve 1) faster than other genomic sites and 2) exhibit a strict molecular clock feature. In our first paper [17], the *GSR* of SARS-CoV-2 was used as $\mu$. We noted that *GSR* is likely lower than $\mu$, because the fraction of near neutral mutation in the whole genome is likely less than 1. Indeed, the substitution rate of 5'UTR of Orf1ab as the fastest rate among all examined segments ($4.3\text{E-}03$ *S/N/Y*) shows good time-independence, and it is 5.3-fold of *GSR*. Therefore, it was set as the lower boundary of $\mu$ in our later studies [20,28], since although the fraction of near neutral mutation in this 5'UTR is much higher than that of the whole genome, it might not reach 1 yet. The upper boundary of $\mu$ were further refined by the lowest $c/\mu$ out of the experiment-determined adaptive mutations from the literature [28]. The true $\mu$ value appears within 1–2.5 fold of the substitution rate of 5'UTR of Orf1ab, which will be used in present study. Encouragingly, this inferred *in vivo* mutation rate falls within the range of the *in vitro* mutation rate measured from cell infection experiments by SARS-CoV-2 [$1.35\text{E-}03$, $20.34\text{E-}02$ *S/N/Y*] with an average rate of $7.49\text{E-}03$ *S/N/Y* [20].

**Step 4E. Calculation of $c/\mu$ from $c$ and $\mu$.** For each genomic segment (*TR*, *UTR*, or *TRS*), we estimated the time-based substitution rate $c$ using linear regression of the cumulative number of substitutions per site over sampling time, as per Step 4B, and the position-based substitution rate ($c$). After calculating $c$ and $\mu$, we then calculated $c/\mu$ for each nucleotide site in the time-based and position-based approaches. These values were then used to generate the discrete and cumulative $c/\mu$ *DFEs* for all 49 segments in SARS-CoV-2.

**Step 4F. Validation and control of potential biases.** Technical and biological bias from our methods and analyzed genomic data could introduce distortions in the $c/\mu$ estimates or generate artifactual L-shaped $c/\mu$ *DFEs*. We employed multiple validation and filtering steps to minimize these biases wherever possible. First, we used SARS-CoV-2 genomes from curated Nextstrain genomic datasets that rigorously filter out sequences with excessive divergence, recombination, missing

data, or known sequencing errors, ensuring sequences are mostly complete and that most observed mutations are mostly from single-point substitutions. Second, we account for population-structure effects by, again, utilizing the NextStrain sequence datasets, which contain both geographically and temporally stratified sequences representing the global variation of SARS-CoV-2 across 19 months. Hence, the resulting L-shaped $c/\mu$ *DFEs* are from the global SARS-CoV-2 variation, thus segment-level *NNBST* versus *NNUST* classifications were indeed stable. We also ensured time-based and position-based $c/\mu$ values were consistent with each other if their segments produced a strict molecular clock feature (see Tables in S2–S5 Tables). The details of this consistency can be found in the results section. In brief. segments exhibiting strict molecular clock showed close agreement in their time-based and position-based $c/\mu$ values, supporting the robustness of the $c/\mu$ calculations, whereas segments not exhibiting strict molecular clocks showed greater deviations in their time-based and position-based $c/\mu$ values.

**Steps 5 and 6. Quantifying the proportions of different mutation types.** The position-based $c/\mu$ for every segment in SARS-CoV-2 was used to generate their respective $c/\mu$ *SSMRRS* (data reported in [28], diagrams for the genome and All-UTR present in Figure of S4 Fig), discrete and cumulative $c/\mu$ *DFE*s. For each segment, all the distributions were L-shaped on the basis that (i) the bin of the highest probability is the lowest $c/\mu$ bin ($c/\mu = 0$) and (ii) subsequent bin frequencies after the most probable bin decrease monotonically. We then defined mutation type classes (strongly deleterious, weakly deleterious, strictly neutral, weakly beneficial, strongly beneficial) by defining three boundaries on the $c/\mu$ axis for the cumulative $c/\mu$ *DFEs*: a weak negative boundary (*WNB*), a neutral boundary (*NB*), and a weak positive boundary (*WPB*), as detailed below. The *WNB*, *NB* and *WPB* boundaries are determined directly from the empirical $c/\mu$ distributions (see Tables 4 and 5 and Table in S8 Table) and not a priori.

The discrete $c/\mu$ *DFE* can precisely determine the fraction of strictly neutral mutations ($f_0 = P(\frac{c}{u} = 1)$) for each segment by simply defining the fraction of mutation sites exhibiting $\frac{c}{u} = 1$. **Eq. 12a** ($\frac{c}{u} = f_0'$) is used to determine the fraction of nearly neutral mutations. From here, the weak negative boundary (*WNB* = $\frac{c}{u} = f_0'$) and the weak positive boundary (*WPB* = 1/*WNB*) with a symmetry assumption can be defined within the cumulative $c/\mu$ *DFE*, where the *WNB* is determined by quantifying its cumulative probability with the function $f_0'$ ($P(x) = \sum_{\frac{c}{u}} P(\frac{c}{u} \geq x) = f_0'$). From the boundaries indicating conserved sites ($\frac{c}{u} = 0$), *WNB* ($\frac{c}{u} = f_0'$), strictly neutral selection ($\frac{c}{u} = 1$) and *WPB* (1/*WNB*), the fractions of strongly deleterious selection ($f^-$, *SN*, $0 \leq \frac{c}{u} <$ *WNB*), weakly deleterious selection ($f_o^-$, *WN*, *WNB* $\leq \frac{c}{u} < 1$), weakly beneficial selection ($f_o^+$, *WP*, $1 \leq \frac{c}{u} <$ *WPB*) and strongly beneficial selection ($f^+$, *SP*, *WPB* $\leq \frac{c}{u} \ll \infty$) can be determined from the cumulative $c/\mu$ *DFE*. One must note that as the *WNB* $c/\mu$ value decreases towards zero, the inferred *WPB* $c/\mu$ rises towards $+\infty$; conversely, as *WNB* increases and approaches $\frac{c}{u} = 1$, *WPB* also decreases and approaches $\frac{c}{u} = 1$, decreasing the boundaries for mutations under weakly neutral selection. The fractions of the four mutation types ($f_o^-$, $f_0$, $f_o^+$, $f^+$), the summation of nearly neutral selection ($f_0' = f_o^- + f_o^+$), the ratios of nearly neutral to strict neutral selection ($f_0':f_0$), weakly deleterious to weakly beneficial selection ($f_o^- : f_o^+$, *WN:WP*) and weakly deleterious to beneficial selection ($f_o^- : (f_o^+ + f^+)$, *WN:P*) were used for validating each segment against the five theories of molecular evolution (see Step 7). We compare the fractions of different mutation types defined using our approach with the conventional approach defined by Ohta [8,31], who proposed the *WNB* and *WPB* $c/\mu$ values as 0.5 and 2, respectively, assuming symmetry. Our $c/\mu$ *DFE* calculations set these parameters for the *WNB* ($\frac{c}{\mu} = 0.52$) and *WPB* ($\frac{c}{\mu} = 2.02$) with respect to the SARS-CoV-2 $c/\mu$ *DFE* data.

**Step 7. Evaluation of SARS-CoV-2 genome and each segment against *ST, KNT, ONNT, NNBST* and *NNUST*.** The presence of the strict molecular clock feature (Yes or No), the abundance of weakly beneficial sites (criteria 1: $f_o^+ \gg 0$), the abundance of weakly deleterious sites (criteria 2: $f_o^- \gg 0$), the abundance of nearly neutral sites (criteria 3: $f_o' \gg 0$), the abundance of weakly and strongly beneficial sites (criteria 4: $f_o^+ + f^+ \gg 0$), and the ratio of nearly neutral to strictly neutral sites (criteria 5: $f_0' \gg f_o$) were used to evaluate each segment against the five evolutionary theories (*KNT, ONNT, ST, NNBST* and *NNUST*) to explain which one(s) most likely explain the segment molecular evolution. The evaluation is a two-step process. First, a segment's applicability to a theory depends on whether its substitution timeline exhibits a strict molecular clock (*KNT* and *NNBST*) or not (*ST, ONNT* and *NNUST*); for example, the SARS-CoV-2 *GSR* exhibits a strict molecular clock, so it will

only be evaluated against *KNT* and *NNBST*. Second, the criteria violations specific to each theory (i.e., predicted fractions of different mutation types and their ratios) are applied to the qualified segment. They are as follows for *KNT* (criteria 3, 4, 5), *ONNT* (criteria 1, 4, 5) and *ST* (criteria 2, 3). To emphasize, genomic segments exhibiting strict molecular clocks and relatively balanced nearly neutral mutation types including *WN* and *WP* were classified as under *NNBST*; conversely, genomic segments not exhibiting strict molecular clocks and imbalanced *WN* and *WP* mutation types were classified as under *NNUST*. As a further robustness check, we repeated the theory evaluation using more conservative Ohta-style boundaries, and *NNST* remained the only theory without violations across the six empirical criteria (see Tables 4–6).

## Significance Statement

Aiming to resolve the long-standing debate between neutral and selectionist theories of molecular evolution can be achieved by developing rigorous, genome-wide tests of selection based on empirical sequence data. Accurately defining the distribution of fitness effects (*DFE*) boundaries separating mutations fixed under genetic drift and natural selection processes can help resolve this debate. Here, we extend a newly developed substitution-to-mutation rate ratio test ($c/\mu$) to 49 functional segments of the SARS-CoV-2 genome. Our findings reveal widespread $c/\mu$ L-shaped *DFEs* and segment-specific deviations from molecular clock assumptions, supporting a unified Near-Neutral Selectionist Theory (*NNST*) that integrates and expands upon classical evolutionary models to incorporate both a Near-Neutral Balanced Selectionist Theory (*NNBST*) and a novel Near-Neutral UnBalanced Selectionist Theory (*NNUST*). This approach offers a framework for quantifying selection strength directly from genomic data and may be broadly applicable across organisms. By mapping well-known empirical phyloP scores to the mutation–selection ratio $\ln(c/\mu)$, our framework enables existing genome-wide phyloP distributions to directly test evolutionary theories, including *NNST*, across deep phylogenetic scales.

## Supporting information

**S1 Table. Nucleotide point mutation data from cell infection experiments.** Reported nucleotide mutations in the cell-based SARS-CoV-2 studies analyzed in our previous study, their locations within the SARS-CoV-2 genome, and the percent substitution rate at these sites computed from our SARS-CoV-2 genomic sequence datasets.
(PDF)

**S2 Table. Linear regression values for non-molecular clock segments Time-based total genomic substitution rate slope and c $R$2 values for each dataset and averaged over each dataset for all SARS-CoV-2 segments not exhibiting strict molecular clock, in order of decreasing average $R$2.** See Figures in S6 and S8 Figs for the timeline slopes of these segments.
(PDF)

**S3 Table. Linear regression values for molecular clock segments.** Time-based total genomic substitution rate slope and c $R^2$ values for each dataset and averaged over each dataset for all SARS-CoV-2 segments exhibiting strict molecular clock, in order of decreasing average $R^2$. See Figures in S7 and S9 Figs for the timeline slopes of these segments.
(PDF)

**S4 Table. Relative rate values for non-molecular clock segments.** Time-based $c/\mu$ ($c/\mu^a$), Position-based $c/\mu$ ($c/\mu^b$), the absolute and percent differences for $c/\mu$ for the genome, All-UTR, All-TR and each coding segment of

SARS-CoV-2 not exhibiting strict molecular clock, in order of decreasing average $R^2$, averaged over the three datasets.
(PDF)

**S5 Table. Relative rate values for molecular clock segments.** Time-based $c/\mu$ ($c/\mu^a$), Position-based $c/\mu$ ($c/\mu^b$), the absolute and percent differences for $c/\mu$ for the genome, All-UTR, All-TR and each coding segment of SARS-CoV-2 exhibiting strict molecular clock, in order of decreasing average $R^2$, averaged over the three datasets.
(PDF)

**S6 Table. Abundance of sites under different selection types for non-molecular clock segments.** Segments are in order of decreasing average $R^2$. (Column 1): Relative abundance of sites under *WN*, *WP* and *SP* selection. (Column 2): Abundance of sites under *NN* and *P* selection. (Column 3): Abundance of sites under *SN*, *NN* and *SP* selection. (Column 4): Abundance of sites under *WN* and *SP* selection. (Column 5): Abundance of sites under *N* and *P* selection. See Fig 4 and Figure of S15 Fig for graphical representations.
(PDF)

**S7 Table. Abundance of sites under different selection types for molecular clock segments.** (Column 1): Relative abundance of sites under *WN*, *WP* and *SP* selection. (Column 2): Abundance of sites under *NN* and *P* selection. (Column 3): Abundance of sites under *SN*, *NN* and *SP* selection. (Column 4): Abundance of sites under *WN* and *SP* selection. (Column 5): Abundance of sites under *N* and *P* selection. See Fig 4 and Figure of S16 Fig for graphical representations.
(PDF)

**S8 Table. Mutation fraction types within the SARS-CoV-2 genome, All-TR and All-UTR.** The boundaries were defined using Ohta's approach versus our approach.
(PDF)

**S9 Table. UTR content, conservation and mutation rates across major life forms.** Percentage of *UTR*, conserved *UTR* and *TR*, synonymous mutations and spontaneous mutation rate in the genome of major life forms and their representative species.
(PDF)

**S1 Fig. Replication-Selection framework and the linked relationship between $c/\mu$ framework and the Mutation-Selection framework and Ka/Ks framework. (A).** Simplified replication-selection model for a virus population over time under positive selection, neutral selection and negative selection for the first nucleotide position (red circle). A detailed description has been placed in the methods section. Viral genomes are shown as black lines. Mutations are shown as red, green and blue circles. **(B).** Integrating $c/\mu$ per nucleotide site framework with MutSel per codon and Ka/Ks per gene frameworks to quantify the transient scaled selection coefficients (S) across different genome regions. $N_e$ represents the effective haploid population size, $P_{fix}$ denotes the probability of fixation, and the fitness of a nucleotide mutant ($F_j$) and wild type ($F_i$) are provided.
(PDF)

**S2 Fig. Genomic computational workflow.** This workflow generates the timelines, *SSMRRS* and *DFE* diagrams, approximating $\mu$, calculating $c/\mu$, determining the proportions of different mutation types and evaluating these mutation proportions against five theories of molecular evolutions.
(PDF)

**S3 Fig. Approximating the true mutation rate ($\mu$) range in a genome and translated coding region. (A)** Genome. **(B)** All-TR. Blue dots represent the $c/\mu$ values at each nucleotide and codon site. Orange dots represent the $c/\mu$ values of top

mutation sites with literature-validated mutation effects. Red dots represent the $c/\mu$ values of conserved sites where lethal mutations for SARS-CoV-2 were experimentally-verified by the literature. Black and purple lines represent the optimized lower and upper boundaries for approximating the true $\mu$ value.
(PDF)

**S4 Fig. Nucleotide position-based $c/\mu$ in the SARS-CoV-2 genome.** (**A**) The structure of the SARS-CoV-2 genome, including major gene segments (Orf1ab, S, E, M and N), accessory genes (Orf3a, Orf6, Orf7a, Orf8 and Orf10), UTRs (Orf1ab 5'-UTR and Orf10 3'-UTR) and TRS (leader TRS-L and TRS-B). (**B**) Position-based $c/\mu$ at each nucleotide site in the SARS-COV-2 genome. (**C**) Position-based $c/\mu$ at each nucleotide site in All-UTR. The red line in S4C Fig represents strict neutral selection ($c/\mu = 1$). $\mu$ is the substitution rate of Orf1ab 5'UTR.
(PDF)

**S5 Fig. Comparison of molecular clocks using different SARS-CoV-2 ancestral sequences.** Percent genomic variation of sequences in set A1a with the references sequences Wuhan-Hu-1 (left) and Wuhan IPBCAMS-WH-01 2019 (right) set.
(PDF)

**S6 Fig. Timelines for non-molecular clock segments across three datasets.** The percent total nucleotide substitution rate for the segments not exhibiting strict molecular clock averaged over the three combined datasets, in order of decreasing average $R^2$ (from left to right, top to bottom). See Table of S2 Table for tabulated regression parameters.
(PDF)

**S7 Fig. Timelines for molecular clock segments across three datasets.** The percent total nucleotide substitution rate for the segments exhibiting strict molecular clock averaged over the three combined datasets, in order of decreasing average $R^2$ (from left to right, top to bottom). See Table of S3 Table for tabulated regression parameters.
(PDF)

**S8 Fig. Timelines for non-molecular clock segments across each dataset.** The percent total nucleotide substitution rate for the segments not exhibiting strict molecular clock for each dataset, in order of decreasing average $R^2$ (from left to right, top to bottom). See Table of S2 Table for tabulated regression parameters.
(PDF)

**S9 Fig. Timelines for molecular clock segments across each dataset.** The percent total nucleotide substitution rate for the segments exhibiting strict molecular clock for each dataset, in order of decreasing average $R^2$ (from left to right, top to bottom). See Table of S3 Table for tabulated regression parameters.
(PDF)

**S10 Fig. Demonstration of NNBST at the subgenetic level of Orf1ab and NSP1–15.** Monthly percent total nucleotide substitution rate for Orf1ab and NSP1–15 genes over 19 months. NSP segments exhibiting faster or slower substitution rates relative to Orf1ab are under weak beneficial and weak negative selection with respect to Orf1ab (effective neutral selection).
(PDF)

**S11 Fig. $c/\mu$ Cumulative probability distribution of non-molecular clock segments.** Segments are in order of decreasing average $R^2$ (from left to right, top to bottom), showing the abundance of sites under different selection types. The boundaries for weak negative (red line) to neutral selection (green line) to weak positive selection (orange line) and their determined $c/\mu$ positions are noted. Strong negative selection and strong positive selection would be to the left and right of the red and orange lines, respectively. A broken line leading on the x-axis represents the distance between

$c/\mu = 3.0$ to the weak positive selection boundary. See Tables 4 and 5 for tabulated $c/\mu$ boundaries and percent abundances for each selection type.
(PDF)

**S12 Fig. $c/\mu$ Cumulative probability distribution of molecular clock segments.** Segments are in order of decreasing average $R^2$ (from left to right, top to bottom), showing the abundance of sites under different selection types. The boundaries for weak negative (red line) to neutral selection (green line) to weak positive selection (orange line) and their determined $c/\mu$ positions are noted. Strong negative selection and strong positive selection would be to the left and right of the red and orange lines, respectively. A broken line leading on the x-axis represents the distance between $c/\mu = 3.0$ to the weak positive selection boundary. See Tables 4 and 5 for tabulated $c/\mu$ boundaries and percent abundances for each selection type.
(PDF)

**S13 Fig. $c/\mu$ Discrete probability distribution of non-molecular clock segments.** Segments are in order of decreasing average $R^2$ (from left to right, top to bottom), showing the abundance of sites under strictly neutral selection. See Tables 4 and 5 for tabulated percent abundances for strictly neutral selection.
(PDF)

**S14 Fig. $c/\mu$ Discrete probability distribution of molecular clock segments.** Segments are in order of decreasing average $R^2$ (from left to right, top to bottom), showing the abundance of sites under strictly neutral selection. See Tables 4 and 5 for tabulated percent abundances for strictly neutral selection.
(PDF)

**S15 Fig. Percent selection types of non-molecular clock segments.** Selection types are decomposed into three categories for strong negative, near-neutral and strong positive selection (top row), weak negative and strong positive selection (middle row) and negative and positive selection (bottom row) using a $c/\mu$ scaling of 5.46 (i.e., lower boundary of true $\mu$). See Table of S6 Table for percent selection type values.
(PDF)

**S16 Fig. Percent selection types of molecular clock segments.** Selection types are decomposed into three categories for strong negative, near-neutral and strong positive selection (top row), weak negative and strong positive selection (middle row) and negative and positive selection (bottom row) using a $c/\mu$ scaling of 5.46 (i.e., lower boundary of true $\mu$). See Table of S7 Table for percent selection type values.
(PDF)

**S17 Fig. phyloP histogram analysis for mammal genomic tracks.** Hg19 phyloP histograms (https://genomewiki.ucsc.edu/index.php/Hg19_phyloP_histograms). Histogram data for phyloP data on the 46-way conservation track on the human genome browser, hg19 each of these data tables have 2,845,303,719 data values. **(A)** Primate subset (primate subset data statistics: minimum: −9.065, maximum: 0.655, mean: 0.04482, standard deviation: 0.600051); **(B)** Placental mammal subset (minimum: −13.796, maximum: 2.941, mean: 0.03594, standard deviation: 0.779426), **(C)**. All 46 vertebrates (minimum: −14.08, maximum: 6.424, mean: 0.0896, standard deviation: 0.833186).
(PDF)

**S1 Data. Meta A1a, A1b, A1c.**
(XLSX)

## Author contributions

**Conceptualization:** Chun Wu.

**Formal analysis:** Nicholas J. Paradis.

**Investigation:** Nicholas J. Paradis.

**Methodology:** Chun Wu.

**Supervision:** Chun Wu.

**Writing – original draft:** Chun Wu, Nicholas J. Paradis.

**Writing – review & editing:** Chun Wu, Nicholas J. Paradis.

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
