## [Decision Letter · Decision Letter 0]

1 Dec 2025

Dear Dr. wu,

Thank you for submitting your manuscript to PLOS ONE. After careful consideration, we feel that it has merit but does not fully meet PLOS ONE’s publication criteria as it currently stands. Therefore, we invite you to submit a revised version of the manuscript that addresses the points raised during the review process.

We look forward to receiving your revised manuscript.

Kind regards,

Ali R. Ansari, Ph.D.

Academic Editor

PLOS ONE

Journal Requirements:

Reviewers' comments:

Reviewer's Responses to Questions

**Comments to the Author**

1. Is the manuscript technically sound, and do the data support the conclusions?

Reviewer #1: Yes

Reviewer #2: Yes

2. Has the statistical analysis been performed appropriately and rigorously?

Reviewer #1: Yes

Reviewer #2: N/A

3. Have the authors made all data underlying the findings in their manuscript fully available?

Reviewer #1: No

Reviewer #2: Yes

4. Is the manuscript presented in an intelligible fashion and written in standard English?

Reviewer #1: No

Reviewer #2: No

Reviewer #1: This study extends a test of selection pressure to 49 segments of the SARS-CoV-2 genome and introduces a unified "Near-Neutral Selectionist Theory" to better explain the virus's evolution

The work is highly original, proposing a new theoretical framework. The manuscript is well-organized.

Minor Improvements:

The abstract notes that 24 segments supported the strict molecular clock (NNBST), while 25 segments did not (NNUST). The Discussion or Results should clearly explain why the UTR/TRS segments appear to have an unbalanced selection mechanism, contrasting the mechanisms driving the balance observed in the translated regions.

Add a simplified, conceptual explanation of what c/mu represents before diving into the population genetics equations.

Ensure consistent formatting of "SARS-CoV-2" (the manuscript sometimes uses "SARS-COV-2").

Reviewer #2: The overall paper is drafted in a form of review report. There are some basic ingredients of a Research Paper is missing. I have some points which can improve the quality of paper:

I. Abstract should be enhanced and can be written in a better way. Add some motivation: what is new?

II. There should be a detailed Conclusion to explain and motivate the readers.

III. The analysis focuses solely on SARS-CoV-2, a rapidly evolving RNA virus; conclusions about NNST may not generalize to organisms with different genomic architectures, population sizes, or evolutionary rates. Please see this carefully.

IV. Lacks methodological details on how c/μ ratios are calculated, validated, or corrected for potential biases like sequencing errors, recombination, or population structure that could artifact the L-shaped DFEs.

V. The criteria distinguishing "L-shaped" DFEs from other shapes and the thresholds separating NNBST from NNUST appear subjective without quantitative cutoffs or statistical validation provided.

VI. SARS-CoV-2 evolution involves immune selection, transmission bottlenecks, and recombination that may independently violate molecular clock assumptions, yet these confounding factors aren't addressed.

**Do you want your identity to be public for this peer review?** For information about this choice, including consent withdrawal, please see our Privacy Policy

Reviewer #1: No

Reviewer #2: No

---

## [Author Response · Author response to Decision Letter 1]

28 Jan 2026

We thank the Editor and Reviewers for their careful reading of our manuscript, “Near Neutral Selectionist Theories (NNST) for SARS-CoV-2 suggested by the substitution–mutation ratio (c/µ) analysis” and for their thoughtful comments.

We have carefully revised the main manuscript and Supporting Information.

Below we provide a point-by-point response to each of the reviewer’s questions.

Reviewer #1

This study extends a test of selection pressure to 49 segments of the SARS-CoV-2 genome and introduces a unified "Near-Neutral Selectionist Theory" to better explain the virus's evolution

The work is highly original, proposing a new theoretical framework. The manuscript is well-organized.

We thank the reviewer for these comments.

Comment 1.

“The abstract notes that 24 segments supported the strict molecular clock (NNBST), while 25 segments did not (NNUST). The Discussion or Results should clearly explain why the UTR/TRS segments appear to have an unbalanced selection mechanism, contrasting the mechanisms driving the balance observed in the translated regions.”

We have added an explanation of the deviating molecular clock presence in the TR vs the UTR/TRS genomic segments, being reflective of their different functional roles, in the Discussion. TRs encode protein that performs various different functions, and nonsynonymous nucleotide substitutions (change protein sequence, lower sequence divergence) are generally less tolerated and are fixed at a slower rate than synonymous nucleotide substitutions (do not change protein sequence, higher sequence divergence) in TRs. The slow nonsynonymous nucleotide substitution rates could be balanced by the faster synonymous nucleotide substitution rates, which may produce the strict molecular clock feature in the total substitution rate. In contrast, UTRs/TRSs have regulatory functions that mainly control protein expression and protein-RNA interactions. Sequence constraint in these genomic segments is rather high, which is not balanced by other mutation types explored in this study. Therefore, an absence of the strict molecular clock feature is often observed in these UTRs/TRSs.

Comment 2.

“Add a simplified, conceptual explanation of what c/µ represents before diving into the population genetics equations.”

We have added a short, plain-language explanation of c/µ in the Introduction, before the technical discussion of the c/µ framework. This paragraph explains that c/µ compares how often mutations arise (µ) to how often they fix (c), and that c/µ > 1, = 1, and < 1 correspond to beneficial, neutral, and deleterious selection, respectively.

Comment 3.

“Ensure consistent formatting of ‘SARS-CoV-2’ (the manuscript sometimes uses ‘SARS-COV-2’).”

We have replaced all occurrences of “SARS-COV-2” and similar variants with the consistent form “SARS-CoV-2” throughout the manuscript.

Reviewer #2:

The overall paper is drafted in a form of review report. There are some basic ingredients of a Research Paper is missing. I have some points which can improve the quality of paper:

I. Abstract should be enhanced and can be written in a better way. Add some motivation: what is new?

We have rewritten the Abstract to better highlight the novelty and motivation of our work. We introduce c/µ in intuitive terms to motivate the elucidation of species fitness using empirical sequence data. We also emphasize the lack of a theoretical framework to explain the non-strict molecular clock behavior of UTR/TRS genomic segments, as per our prior report.

II. There should be a detailed Conclusion to explain and motivate the readers.

We already provide a Conclusion section following the Discussion. We also added a concluding sentence urging the re-examination of intra-species molecular evolution using available empirical sequence data to highlight that a blend of genetic drifting, natural selection and nearly neutral selection contribute to species molecular evolution.

III. The analysis focuses solely on SARS-CoV-2, a rapidly evolving RNA virus; conclusions about NNST may not generalize to organisms with different genomic architectures, population sizes, or evolutionary rates. Please see this carefully.

In response to the reviewer’s concern that NNST may be applicable only to SARS-CoV-2, we emphasize that our framework is explicitly grounded in, and supported by, genome-scale comparative evidence far beyond viral evolution. phyloP is a well-established comparative-genomics method that quantifies deviations from neutral evolution across dozens to hundreds of species, from primates to all vertebrates. Critically, the phyloP score −ln(r) is mathematically equivalent (up to sign convention) to ln(c/μ), allowing existing phyloP genome-wide distributions to be reinterpreted directly within a mutation–selection framework. Analysis of the classic 46-species phyloP histograms—covering 2.85 billion nucleotide positions—reveals a robust, recurring pattern across evolutionary scales: a dominant near-neutral mode, persistent right-skew toward conservation, and a non-vanishing accelerated tail. This structure is inconsistent with strict Kimura Neutral Theory (which predicts centering at zero after sufficient divergence) and with strong selectionist models (which would suppress the near-neutral peak), but is exactly the qualitative signature predicted by NNST. Moreover, the scale dependence of the distributions—L-shaped at shallow divergence (mirroring rapidly evolving systems like SARS-CoV-2), broadening yet remaining biased at deep phylogenetic scales—demonstrates that near-neutral balance is not a virus-specific phenomenon, but a general property of genome evolution. Thus, rather than being limited to SARS-CoV-2, NNST provides a unifying theoretical interpretation for empirical patterns already observed across vertebrate and mammalian genomes, as independently captured by decades of phyloP analyses.

IV. Lacks methodological details on how c/μ ratios are calculated, validated, or corrected for potential biases like sequencing errors, recombination, or population structure that could artifact the L-shaped DFEs.

We have added a “Calculation of c, µ, and c/µ” subsection specifying how c is obtained from time-resolved substitution timelines and how µ is approximated using the effective neutral substitution rate, in the Methods. We added a “Validation and bias control” subsection describing how we use Nextstrain’s curated datasets and additional site/segment filters to limit sequencing artifacts, exclude instances of recombination and compare time-based and position-based c/µ estimates as an internal consistency check. We explicitly link these steps to the L-shaped DFEs and emphasize that the observed L-shapes persist after these controls, indicating the L-shaped DFEs are not likely artificially produced.

V. The criteria distinguishing "L-shaped" DFEs from other shapes and the thresholds separating NNBST from NNUST appear subjective without quantitative cutoffs or statistical validation provided.

We have clarified and quantified how DFE shapes and theoretical regimes are defined in the Methods. Specifically, we have included a plain-language description on defining the L-shaped c/u DFE and in defining the three boundary points in quantifying the different mutation types in Step 5 and 6 of the Methods section. A more technical description of defining these information, as well as the comparison of our approach to that defined by Ohta, is continued in the following paragraph.

VI. SARS-CoV-2 evolution involves immune selection, transmission bottlenecks, and recombination that may independently violate molecular clock assumptions, yet these confounding factors aren't addressed.

We have included a paragraph in the Discussion of how the differing biological processes (immune selection, transmission bottlenecks, and recombination) could violate the molecular clock feature and how these processes are either absent or present in the current SARS-CoV-2 genomic dataset analyzed in this study. In brief, we outline that the impact of immune selection on key SARS-CoV-2 genomic segments (i.e. ORF6 and ORF7a proteins) and their absent molecular clock feature could be explained by host immune-driven selection. The impact of transmission bottle necks on the molecular clock was just outlined. Conversely, the impact of recombination on our SARS-CoV-2 genomic dataset is likely minimal since it does not contain viral sequences from the Omicron XBB or Deltacron lineages.

Comments to the Author

1. Is the manuscript technically sound, and do the data support the conclusions?

Reviewer #1: Yes

Reviewer #2: Yes

2. Has the statistical analysis been performed appropriately and rigorously?

Reviewer #1: Yes

Reviewer #2: N/A

3. Have the authors made all data underlying the findings in their manuscript fully available?

Reviewer #1: No

Reviewer #2: Yes

4. Is the manuscript presented in an intelligible fashion and written in standard English?

Reviewer #1: No

Reviewer #2: No

5. Review Comments to the Author

6. PLOS authors have the option to publish the peer review history of their article (what does this mean?). If published, this will include your full peer review and any attached files.

Do you want your identity to be public for this peer review? For information about this choice, including consent withdrawal, please see our Privacy Policy.

Reviewer #1: No

Reviewer #2: No

---

## [Editor Report · Decision Letter 1]

5 Feb 2026

Near Neutral Selectionist Theories (NNST) for SARS-COV-2 suggested by the substitution-mutation ratio (c/µ) analysis

PONE-D-25-58778R1

Dear Dr. wu,

We’re pleased to inform you that your manuscript has been judged scientifically suitable for publication and will be formally accepted for publication once it meets all outstanding technical requirements.

Kind regards,

Ali R. Ansari, Ph.D.

Academic Editor

PLOS One
---

## [Editor Report · Acceptance letter]

PONE-D-25-58778R1

PLOS One

Dear Dr. Wu,

I'm pleased to inform you that your manuscript has been deemed suitable for publication in PLOS One. Congratulations! Your manuscript is now being handed over to our production team.

Kind regards,

on behalf of

Prof. Ali R. Ansari

Academic Editor

PLOS One